# Detecting changes in dynamic and complex acoustic environments

Yves Boubenec[1,2*†], Jennifer Lawlor[1,2†], Urszula Górska[3,4,5], Shihab Shamma[1,2,6,7], Bernhard Englitz[1,2,3]

[1]Laboratoire des Systèmes Perceptifs, CNRS UMR 8248, Paris, France; [2]Département d'études cognitives, École normale supérieure, PSL Research University, Paris, France; [3]Department of Neurophysiology, Donders Centre for Neuroscience, Radboud Universiteit, Nijmegen, Netherlands; [4]Psychophysiology Laboratory, Institute of Psychology, Jagiellonian University, Krakow, Poland; [5]Smoluchowski Institute of Physics, Jagiellonian University, Krakow, Poland; [6]Department of Electrical and Computer Engineering, University of Maryland, College Park, United States; [7]Institute for Systems Research, University of Maryland, College Park, United States

*For correspondence:
boubenec@ens.fr

[†]These authors contributed equally to this work

Competing interests: The authors declare that no competing interests exist.

**Abstract** Natural sounds such as wind or rain, are characterized by the statistical occurrence of their constituents. Despite their complexity, listeners readily detect changes in these contexts. We here address the neural basis of statistical decision-making using a combination of psychophysics, EEG and modelling. In a texture-based, change-detection paradigm, human performance and reaction times improved with longer pre-change exposure, consistent with improved estimation of baseline statistics. Change-locked and decision-related EEG responses were found in a centro-parietal scalp location, whose slope depended on change size, consistent with sensory evidence accumulation. The potential's amplitude scaled with the duration of pre-change exposure, suggesting a time-dependent decision threshold. Auditory cortex-related potentials showed no response to the change. A dual timescale, statistical estimation model accounted for subjects' performance. Furthermore, a decision-augmented auditory cortex model accounted for performance and reaction times, suggesting that the primary cortical representation requires little post-processing to enable change-detection in complex acoustic environments.

## Introduction

Many natural and environmental sounds are composed of shorter, elementary events, whose occurrence can be described on a statistical level (*Lederman, 1979*; *McDermott and Simoncelli, 2011*; *Thoret et al., 2014*; *Turner and Sahani, 2007*). For example, individual drops of water can add together to sound like rain or like a dripping faucet, depending on their number, rate, and relative timing (*McDermott et al., 2013*). However, in real-life, listeners face a dynamic acoustic environment, where statistics do not remain constant for very long. Changes in the statistics of the sound of rustling leaves amidst the sounds of an ongoing storm, or changes in the acoustic composition of a busy cityscape, provide relevant information of putative threats. We investigate here determinants of human performance and their neural representation in these contexts, addressing the hypothesis that the behavior and neural representation are consistent with statistical estimation.

Changes in sound statistics can only be detected if the statistical properties before the change have been estimated sufficiently well (*Kaya and Elhilali, 2014*; *McDermott et al., 2013*). Without this estimate, the listener cannot distinguish between 'what to ignore' given the current statistics

and 'what to recognize' as a change. Moreover, the quality of this estimate can influence the speed and certainty of detection, which are essential in real-life contexts. The present study thus investigates the factors influencing detection of deviations in sound statistics, and what the underlying dynamics of auditory sensory and evidence accumulation processes are in the human brain. For this purpose, listeners are presented with a continuous sound, whose statistics change at a random time. Hence, they are faced with the dual-task of estimating the baseline statistics and detecting a potential change in those statistics at any moment, which mimics real-life challenges.

The estimation of sound statistics depends on many factors, but most importantly on the complexity of a stimulus in relation to the time available to sample it (*Kaya and Elhilali, 2014*). A simple stimulus, governed by only few parameters, can be reliably estimated more quickly than a complex stimulus. We introduce a statistically controlled stimulus that combines simplicity with broad spectral distribution. In contrast to previous studies with narrow-band complex stimuli (*Andreou et al., 2015*; *Cervantes Constantino et al., 2012*; *Overath et al., 2010*; *Teki et al., 2013*), the sounds here form a minimalistic, but well-controlled model for natural, acoustic textures that are only defined by first order statistics. The task for the subjects was to listen to the texture of the stimulus (for a variable pre-change duration), and then signal the detection of a change in the texture as soon as possible.

We found that detection performance improves with the time available to sample the baseline statistics before the change. As expected, detection performance also depended on the saliency of the change. EEG recordings from auditory projection sites show a strong response related to the onset of the sound, but did not exhibit a discernible response related specifically to the subsequent change in stimulus statistics. By contrast, EEG signals over parietal cortex appeared after the time of change, and displayed a build-up rate that depended on the size of the change (consistent with EEG responses in other evidence integration tasks, e.g. *O'Connell et al., 2012*; *Kelly and O'Connell, 2013*). The peak amplitude of this potential also increased with change size, but decreased with pre-change interval, *i.e.* the time available to the subjects to sample the stimulus baseline statistics. Performance and reaction times were well predicted by a model of statistical estimation based on the difference in the outputs of two leaky integrators operating at fast and slow timescales. In addition, a model of auditory cortical processing (*Chi et al., 2005*; *Overath et al., 2008*) augmented with an accumulation-to-bound decision stage also accounted for the EEG responses and subjects' behaviors, thus suggesting that decision-making in such statistically complex acoustic environments may only require minor post-processing (channel-selection and averaging) beyond the early auditory cortex.

## Results

We investigated the neural mechanisms of detecting changes in the statistics of auditory stimuli, on the basis of human behavioral performance, neural response and models of acoustic processing leading to decision-making. In a set of psychoacoustic experiments, listeners (n = 12) were presented with complex acoustic stimuli, whose statistics could change at a random time. Several parameters of the change were varied in order to estimate their influence on the change's saliency. In a different set of listeners (n = 18) EEG responses were collected to track the brain dynamics reflecting the accumulation of sensory evidence leading to the detection of a change in sound statistics. We propose a simple model to account for the listener's behavior, which is based on the estimation of stimulus statistics on two timescales. Finally, we suggest a neural implementation of this principle based on a model of auditory cortical processing.

### Detection of changes in statistics is consistent with estimation of marginal distribution

The ability to detect a change in stimulus statistics improved in trials that provided more time before the change ('change time' in *Figure 1A*) for subjects to listen to the baseline statistics of the texture. Performance also increased monotonically to different asymptotic levels for the four tested change sizes (50, 80, 110, 140%, *Figure 2A*). Asymptotic performance depended on change size, with bigger changes in marginal probability leading to greater asymptotic performance especially between levels, from 50% to 95% (*Figure 2A*, $p_{size} < 10^{-5}$, Friedman; $p_{time} < 10^{-5}$, Friedman). Change size also influenced the dependence on change time, such that greater change sizes led to improved

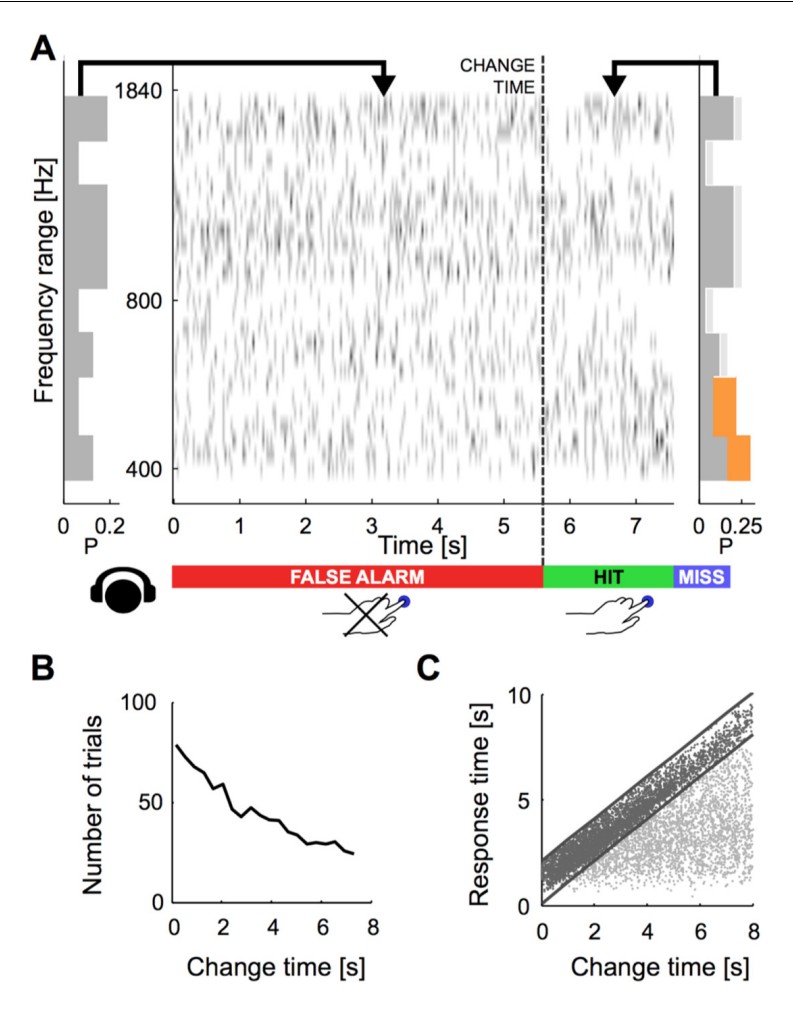

**Figure 1.** Dynamical change-detection paradigm with auditory textures. (**A**) Subjects listened to an acoustic textural stimulus, whose predictability was governed by its marginal frequency distribution (grey curve, left panel). Tones in individual frequency bins were drawn independently consistent with the marginal (middle panel). Listeners were instructed to report changes by a button press. The frequency marginal was modified (indicated in orange in the right panel distribution) after a randomly chosen point in time ('change time'). The probabilities in two adjacent or non-adjacent frequency bins were increased together, and the distribution over the bins renormalized to maintain average global level. (**B**) The distribution of change times was chosen from an exponential distribution. This ensured that the probability of a change in the next time-bin remained constant (shown here is the empirical distribution). (**C**) Response times occurred before (false alarms) and after the change time (hits). Subjects usually responded only after an initial listening duration, allowing them to acquire the sound statistics.

performance at shorter change times than for smaller change sizes (*Figure 2A*). This translates to a combined steepening and leftward shift of the performance curves with change size. The significance of this effect was assessed by fitting the performance curves for individual subjects with a parametric function of sigmoidal shape (an Erlang CDF, see Materials and methods) in order to extract the change size-dependent time constant. The characteristic time constant τ decreased significantly as a function of change size (*Figure 2B*; $p < 10^{-6}$, Kruskal-Wallis).

The observed performance could alternatively be explained by a timing strategy or a pattern recognition strategy. Both of these explanations can be rejected based on the data and the paradigm: if subjects had used a timing strategy, their instantaneous false alarm rate (as a function of change time, divided by the window length) should never reach a constant value. Instead, the false alarm

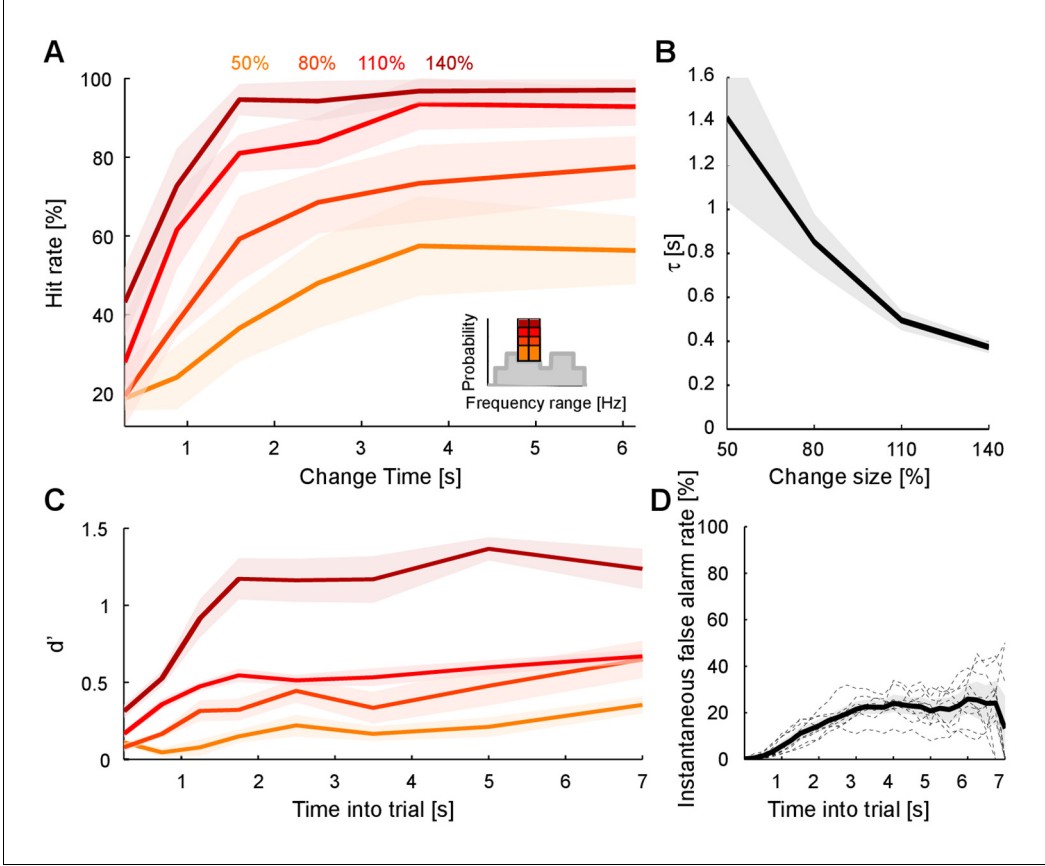

**Figure 2.** Detecting a change in statistics improves with size and time of change. (A) Performance of change detection depended significantly on change time (abscissa) and change size (shades of orange indicate the step size as percent of the original bin probability, see inset). Only changes in contiguous bins were used presently, to maintain identical trial numbers across difficulties. (B) The dynamics of the performance curve varied with change size, indicated by the speed parameter $\tau$ of an Erlang CDF fitted to the data (see Materials and methods). (C) Dynamical d' confirms the dependence of performance on change time and change size. The dependence on change time suggests an improved detection relying on a converged estimate of the baseline statistics, whereas the dependence on change size indicates a higher level of certainty can be attained more rapidly if the amount of evidence is larger. (D) Instantaneous false alarm rate is uniform across time, after an initial hesitation to respond in the first 2 s. The initial hesitation is likely due to the task-design, requiring an initial estimation of the sound statistics.

The following figure supplements are available for figure 2:

**Figure supplement 1.** Change detection improves with base probability.

**Figure supplement 2.** Change detection is not focussed on high probability bins.

**Figure supplement 3.** Change detection improves with stimulus exposure in the previous trial.

rate exhibits an initial linear increase, followed by a constant false alarm rate per unit time (*Figure 2D*), a feature that was embodied in the behavior of the models (see Figures 7E/8F). Furthermore, the initial rising portion of the false alarm rate is a consequence of the dual estimation task design. The uniform regime of false alarm rate is consistent with the use of an exponential distribution of change times, which keeps the change occurrence probability constant per unit of time (see *Figure 1B* and Materials and methods).

Some subjects could have attempted to use a pattern recognition strategy, i.e. effectively ignoring the statistics of the first stimulus. However, based on the stimulus design, a pattern recognition

strategy would have failed, since the first stimulus was drawn randomly for each trial, and the second was a stochastic modification of the first. Further, in this case, detection performance should not have depended on change time. All together these results are inconsistent with both a pattern recognition and a timing strategy.

Using the time-dependent false alarm rate, the sensitivity of the subjects to detect a change can be analyzed with a time-dependent d' (see Materials and methods for details of computing this d'). This analysis exhibited similar monotonically increasing shapes as a function of both change time and size (*Figure 2C*). Further, probability in a frequency bin was positively correlated with change detection (*Figure 2—figure supplement 1*), consistent with the idea that a high rate of samples provided a better estimate of the probability value in a frequency bin. We can rule out that only large probability bins were attended to, since the performance for equal size chances in large probability bins is dominated by the change in other, lower probability bins (*Figure 2—figure supplement 2*). Finally, longer stimulus duration in the current trial predicted a reduced performance in the following trial (*Figure 2—figure supplement 3*), suggesting that the converged estimate in the previous trial could 'contaminate' the estimation process in the subsequent trial. This is another indication that subjects were not using a pattern recognition strategy, as such a strategy completely ignores the statistics presented in the previous trial.

In summary, those findings indicate that change detection (i) improves with time allowed to sample the stimulus, (ii) improves with the size of the change and (iii) saturates with longer observation intervals. These properties are consistent with statistical decision-making, where a decision can only be made if the observed change in a stimulus property is substantial compared to the current uncertainty about the same property. Subjects using statistical decision making can (i) reduce their uncertainty by collecting more stimulus information over time, (ii) use larger differences in the stimulus property to overcome the uncertainty more rapidly, and (iii) will not be able to improve their performance once the estimation of the stimulus statistics has saturated.

## Reaction Times are consistent with statistical estimation

The dependence of performance on *change time* suggests a dynamical mechanism performing an on-going estimation of the initial statistics. To gain insights into these dynamics, we examined the dependence of reaction times on the parameters of the change, especially its size, which intuitively correlates inversely with task difficulty according to Piéron's law (*Pins and Bonnet, 1996*) and time of occurrence (or 'change time').

Reaction time distributions changed both in duration and shape as a function of change size (*Figure 3A*). Median reaction time decreased with larger change sizes ($p<10^{-3}$; Kruskal-Wallis, *Figure 3B*), in accordance with the increase in performance with larger change sizes. Receiver operating curve (ROC)-based analysis indicated that the distributions of reaction times were different across change sizes and chance level (*Figure 3—figure supplement 1*; $p<10^{-7}$; Friedman). More specifically we found a significant difference between the most difficult condition and chance level ($p<10^{-5}$; Kruskal-Wallis), confirming that subjects were performing at all change sizes. This suggests that the time necessary to detect the deviation between the pre- and post-change stimulus statistics was reduced for larger change sizes.

For shorter change times, reaction time distribution changed in a qualitatively similar manner as was observed for smaller change sizes, although the effect was less pronounced (*Figure 3C*). Median reaction times decreased with change times, mirroring dependence of performance on change times ($p<10^{-5}$; Kruskal-Wallis, *Figure 3D*). This dependence can already be seen in the raw data (*Figure 1C*), where hit trials (black) for longer change times exhibited shorter reaction times. Again the timing of the first correct responses decreased correspondingly with longer change time, suggesting more accurate estimation of the initial statistics.

## Dependence on spectral location of acoustic change

Changes in stimulus statistics are effectively a probabilistic redistribution of the stimulus energy in the spectrotemporal domain, here restricted to the spectral axis. Therefore, we hypothesized that a detection process acting in a spectrally localized manner should perform better when the total energy of the change is concentrated in a restricted frequency region. Indeed, we found that performance decreased for non-localized changes when compared with localized ones (*Figure 4A*). This

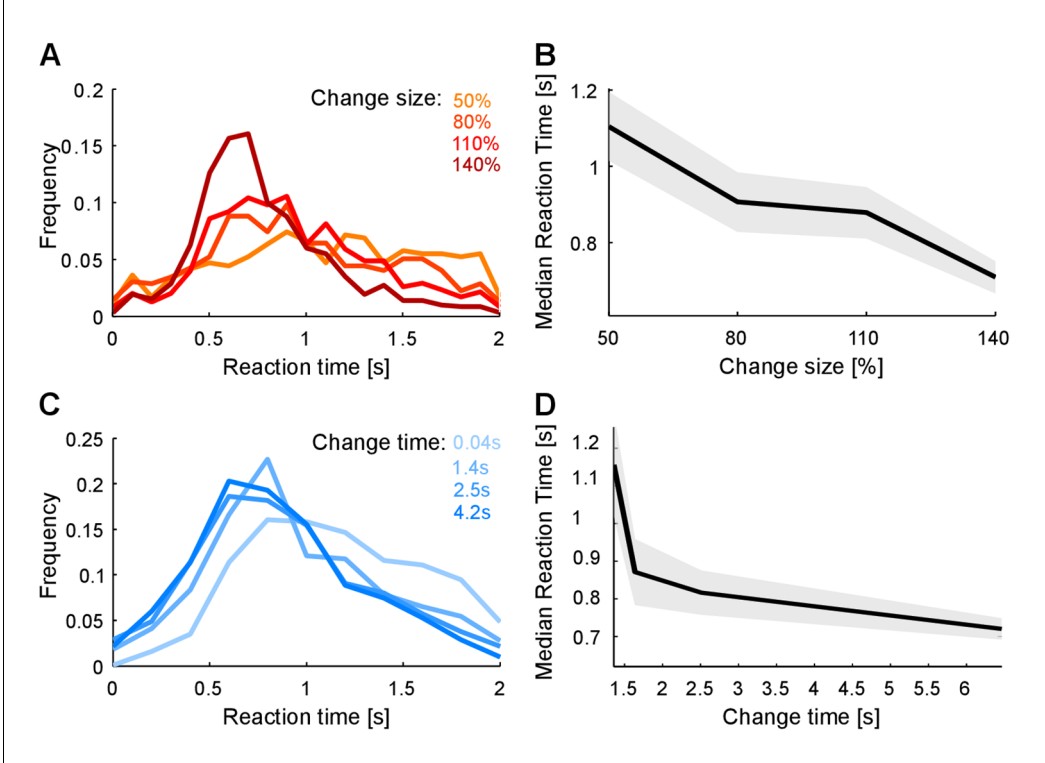

**Figure 3.** Reaction times also reflect estimation of pre- and post-change stimulus properties. (**A**) Reaction time distribution sharpens with change size. (**B**) Median response time significantly reduces by 20% (p<10$^{-4}$, Kruskal-Wallis) with larger change size (different colors indicate different change sizes). These effects indicate a faster, temporally more constrained decision, which could indicate more rapid evidence accumulation for larger changes. (**C**) Reaction time distribution sharpens with change time and **D**) median reaction time reduces rapidly with change time by 25% (p<10$^{-5}$, Kruskal-Wallis). Both effects indicate a higher degree of certainty in decision making, which could indicate a more converged estimation of the pre-change statistics.

The following figure supplement is available for figure 3:

**Figure supplement 1.** Discriminative performance across change sizes.

effect was significant only when distances below and above eight semitones were grouped (*Figure 4A*; p<5.10$^{-3}$). Finally, we found that performance did not vary as a function of relative position along the frequency axis (p=0.28; *Figure 4B*), contrary to the predictions of a recent study (*Catz and Noreña, 2013*) showing that the cortical representation at the extreme edges of the stimulus spectrum could be enhanced for sharp contrast, resulting in lower change detection thresholds.

## EEG responses correlate with accumulation of sensory evidence

We collected neural responses using electroencephalography (EEG) in human subjects performing the above psychoacoustic task to study the relationship between behavioral performance and neural responses, and to narrow down the scalp regions whose neural response reflects the change in statistics. The analysis was focused on a subset of the recording electrodes, namely an auditory (central location, El.1; corresponding to the center in *Nie et al., 2014*) and a centro-parietal (14,27,28; corresponding to *Twomey et al., 2015*) set. Depicted potentials show averages across each set of electrodes. Subjects exhibited similar performance and reaction time dependencies on change time as in the psychophysical experiments (*Figure 5—figure supplement 1*). Change times were binned into four bins based on their distribution and Hit rate to equalize trials per bin.

At stimulus onset, the average auditory potential exhibited a classical, large and rapid event-related potential (ERP) (*Figure 5A,C*, composed of N1 and P2), followed by a negative sustained

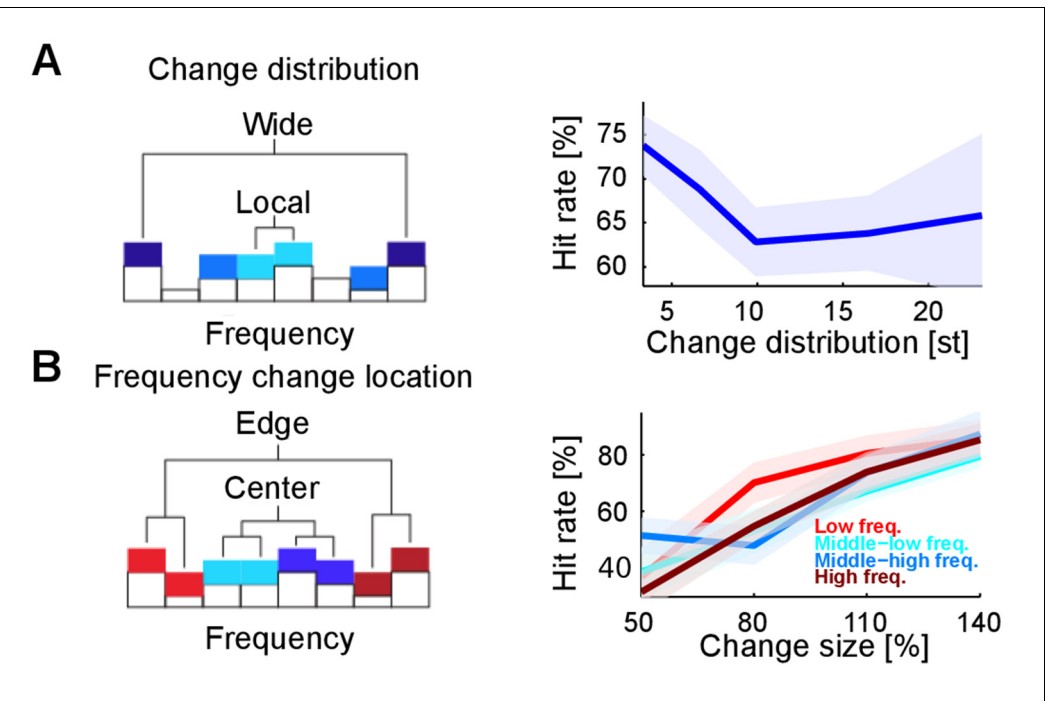

**Figure 4.** Detectability of changes depends on spectral properties of the change. (**A**) Spectral distance between the changed bin centers ('change distribution', measured in semitones, st) significantly reduces performance (p=0.01, Kruskal-Wallis test). Spectral distance ranged from neighboring (three st) bin centers to locations at the edges of the tested range (23 st). (**B**) Absolute spectral position of the changed bins does not influence performance (p=0.85, Kruskal-Wallis). Absolute spectral position was not significantly correlated with the detectability.

potential (indicated as NS in the figure) previously described for prolonged stimulus duration (*Hari et al., 1980*; *Lammertmann and Lütkenhöner, 2001*; *Lütkenhöner et al., 2011*). However, there was no systematic evidence for a response to the change in statistics (*Figure 5B1*, EEG of Hit trials aligned to change times). EEG aligned to subjects' response time also did not show a significant response (*Figure 5B2*, EEG of Hit trials aligned to button-press, different colors indicate different change sizes, averaged over all change times, see below for differences in change time). This suggests that the detection of the change in statistics was not accompanied by an overall response in the auditory cortex comparable to other stimulus changes such as stimulus onset or offset (compare also to the model responses in Figure 8B, see also Discussion). While this does not preclude the information about the change to be available in early auditory cortex, there is no specific, *overall* reaction to the change, compared to the continuous representation of the stimulus.

The centro-parietal electrodes exhibited a centro-parietal positivity (CPP) reported previously (*O'Connell et al., 2012*; *Kelly and O'Connell, 2013*; *Twomey et al., 2015*) in a similar location (see *Figure 5F* for its topography at response time). In contrast to the central electrodes, the CPP did not display any clear response to sound onset (*Figure 5D*) but exhibited a long-lasting response following change events (*Figure 5E1*). This increase in the EEG signal was building-up and preceded subjects' responses across change sizes (*Figure 5E2*), outlasting the timing of the button press. The difference between change sizes was colocalized with the CPP (*Figure 5F*, inset), indicating that the difference in amplitude is not due to a global shift in potential). In previous studies, the CPP potential was clearly linked to evidence integration in decision making tasks, e.g. in simple visual and auditory detection tasks (*O'Connell et al., 2012*) and a complex visual discrimination task (*Kelly and O'Connell, 2013*). We therefore hypothesized the CPP to also be indicative of evidence integration in complex auditory detection tasks. In order to assess this, we examined how the CPP potential depended on the amount of evidence, and whether it exhibited accumulation-to-threshold dynamics.

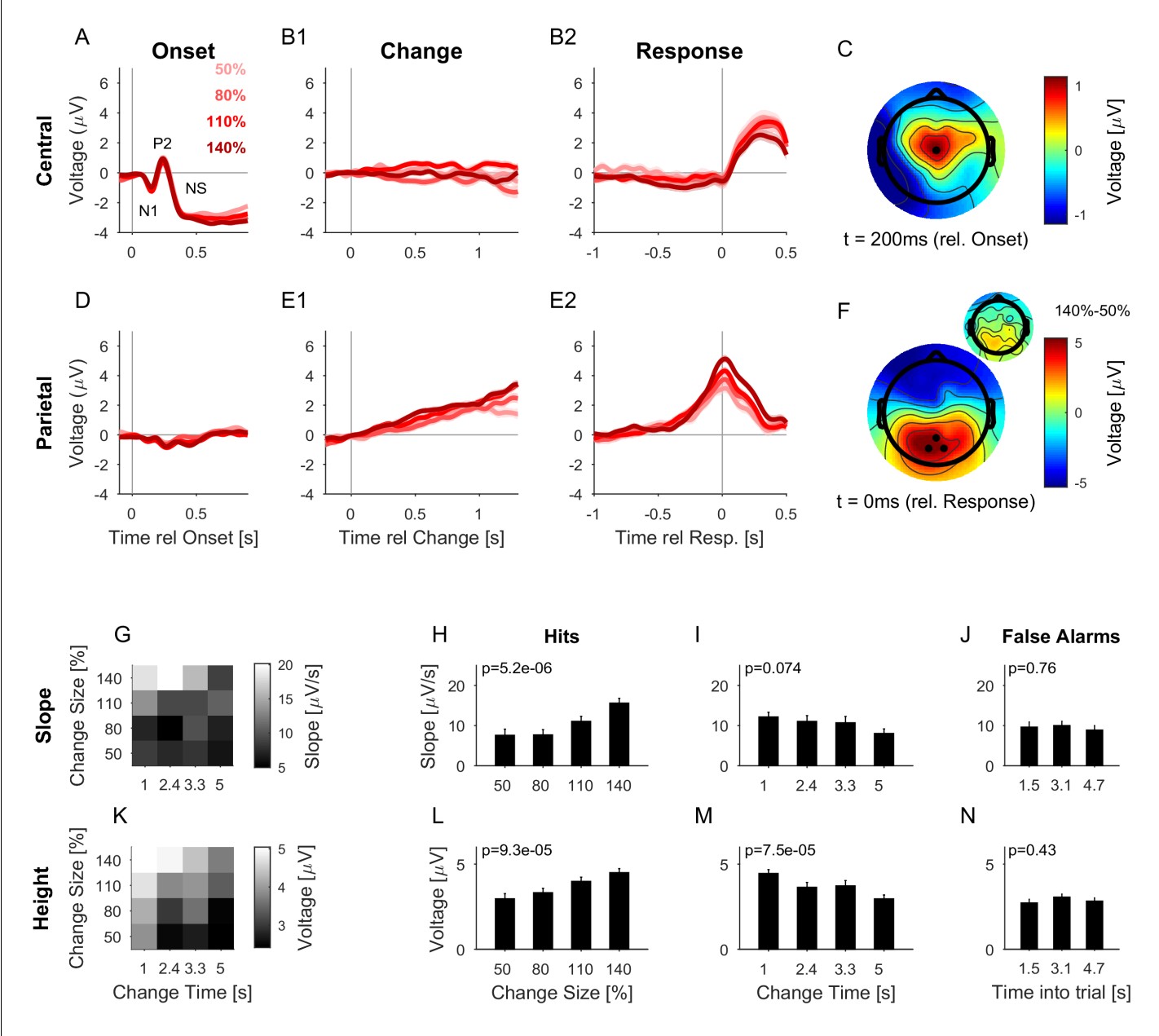

**Figure 5.** The CPP potential shows a dependence on both time and size of change, while the central potential remains unaffected. (A) After stimulus onset, the central potential (Ch. 1, black dot in C) shows a classical N1-P2 progression, followed by a sustained negative potential (labelled NS here). Different shades of red indicate different change sizes. Curves are average over all change times, to avoid crowding the plots. Note that the lowpass filtering at 20 Hz (common for all potentials) reduces the N1/P2 amplitudes below their typical size. (B1) Locked to the time of change, the central potential shows a slow negative trend, which, however, does not depend systematically on change size. (B2) Preceding the response, the central electrodes show no significant change in potential, which only starts to deviate from 0 after the button press. (C) At 200 ms after stimulus onset, the topography of the potential indicates a typical auditory onset response for bilateral stimulation, i.e. centered on Cz (El.1 in the equidistant layout, black dot). (D) The potential above the central parietal cortex (average over Ch. 14,27,28 in the equidistant cap, black dots in F) shows no substantial change at stimulus onset. (E1) Aligned to the time of change, the CPP electrodes show a progressive increase in potential, with some staggering according to change size. In comparison to the response-locked potentials, the present potential is wider and smaller since it is composed of responses at different times. (E2) In contrast to the central electrodes, the CPP electrodes show a clear increase before the response, peaking at or slightly after the response time. (F) The topography locked to the response is found to be centered over the parietal cortex, tending towards the occipital cortex (black dots mark Ch. 14,27,28). The inset shows the difference between the 140% and 50% condition, indicating that the difference in potential is also localized consistently with the average topography. Note, that there was no display change in the entire tone presentation, and a 0.5 s gap after the response, before the screen changed, hence, visual responses can be excluded. (G) CPP slope of the potential leading up to the response in relation to the

*Figure 5 continued on next page*

*Figure 5 continued*

different change time and size conditions was measured in a window of 300–50 ms before the response. (H) CPP slope depended significantly on change size (2-way ANOVA with change time and change size as factors, p<<0.001 for the change time as a factor). (I) CPP slope did not depend significantly on change time (ANOVA as above, p=0.07). (J) CPP slope for false alarms showed no significant dependence on the time into the trial (p=0.76, 1-way ANOVA). (K) Peak height of the CPP was measured in a symmetric window of 80 ms around the response time. (L) Peak height of the CPP showed a significant increase with change size (2-way ANOVA with change time and size as factors, p<<0.001 for change size). (M) Peak height depended significantly on change time, decreasing with longer change times (ANOVA as above, p<<0.001 for change time). (N) Peak heights for false alarms showed no dependence on time into the trial (p=0.43, 1-way ANOVA) but were significantly smaller than the hit trials (p<1e-9, 1-way ANOVA). Error bars indicate single SEMs for all plots.

The following figure supplements are available for figure 5:

**Figure supplement 1.** Change detection performance during the EEG experiment.

**Figure supplement 2.** Same data and analysis as in *Figure 5*, however, detrended with a classical high-pass filter (Matlab: *filtfilt*, 0.1 Hz, 15th order, 50 dB attenuation in the stop band).

Both the slope (*Figure 5G*) and the height (*Figure 5K*) of the response-aligned CPP potential depended on the stimulus parameters. The slope increased significantly with change size (*Figure 5H*, p<<0.001, 2-way ANOVA across change size and change time), but was not significantly dependent on change time (*Figure 5I*, p=0.074, same ANOVA). The effect of change size on slope is consistent with a representation of task-related evidence in the CPP signal, as reported previously in other change detection tasks (*O'Connell et al., 2012*).

The height of the potential also increased significantly with change size (*Figure 5L*, p<<0.001, 2-way ANOVA across change size and change time), and decreased significantly as a function of time (*Figure 5M*, p<<0.001, same ANOVA). Such a change size dependence has been reported before (see *Figure 2* in *O'Connell et al., 2012*), and at first appears inconsistent with a fixed threshold. However, since the execution of the button press requires some time, the application of the threshold has to precede the button press by some delay. The observed difference in heights could thus reflect a continued accumulation of evidence at different slopes, during the time interval between decision commitment and response completion, until the execution of the decision is communicated to the CPP source. Consistent with this interpretation, CPP height did not exhibit a dependence on change size, if measured in a window of 200–100 ms preceding the response time (p=0.16, same ANOVA, close to the crossing in *Figure 5E2*). In addition, we verified that the CPP height did not depend on the reaction time (*Figure 6*), as expected from an evidence accumulator signal (*Kelly and O'Connell, 2013*).

The height decrease as a function of change time is indicative of a reduction in threshold as a function of time (*Figure 5M*). However, we did not observe an increase in FA rate later in the trial (*Figure 2D*), suggesting no increase in unfounded decisions. Although the time-dependence of CPP height could result in a decrease of CPP height for late versus early reaction times, we did not find any significant decrease in CPP height for late reaction times, which may be due to a rather small effect-size (*Figure 6B*).

Finally, CPP responses aligned to false alarms exhibited similar slope and amplitude as the lower signal conditions (50%, 80%), however, were overall significantly lower than the overall signal conditions (p<<0.001, 1-way ANOVA, across change size). Neither slope nor height displayed a dependence as a function of time into the trial (*Figure 5J/N*, p=0.76, p=0.43, respectively, 1-way ANOVA, across different time-into-trial bins). Together these results suggest that the decision threshold on the CPP is close to the lowest change size / false alarm height.

Neither of these results depended on the detrending method, as verified by the alternative use of a classical high-pass filter (see Materials and methods and *Figure 5—figure supplement 2*).

In summary, we found central and centro-parietal electrodes to respond in a diametrically opposed manner to stimulus onset and (detection of) change in statistics. The CPP potential remained practically silent to stimulus onset, but reflected properties of the stimulus/task when aligned to button press. These results reinforce the notion that the CPP potential reflects sensory evidence accumulation and exhibits accumulation-to-threshold dynamics, with the possibility of

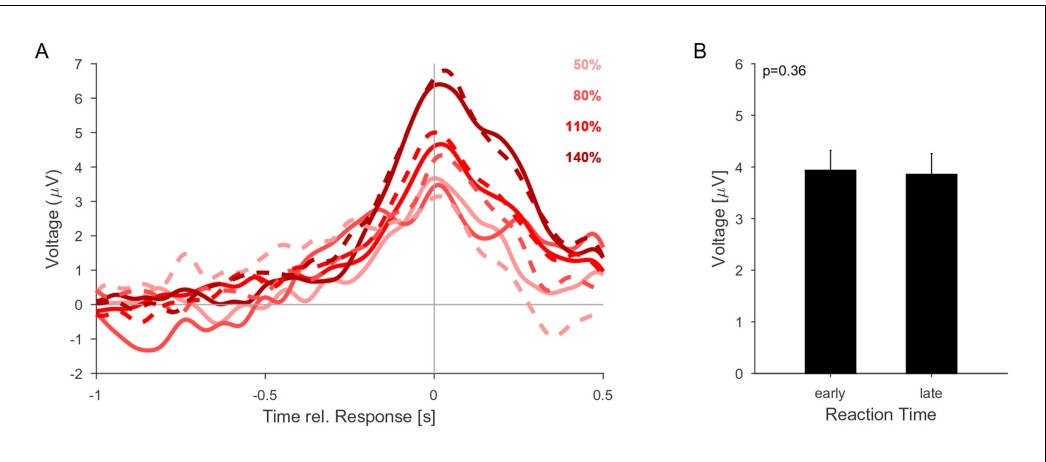

**Figure 6.** The CPP potential shows no dependence on whether responses occur early or late after the change. (**A**) CPP potentials aligned to response as in *Figure 5E2* (for second change-time bin, i.e. around 2.4 s). The solid lines are the early responses (up to median reaction time) and the dashed lines are the late responses (median reaction time to end of response-window). (**B**) Across all conditions the reaction time did not significantly influence the height of the CPP potential (p=0.36 for reaction time, 3-way ANOVA over reaction time, change size and change time).

continued integration until actual response execution. As a function of change time, only the CPP potential's height reduced, suggesting a time-dependent threshold.

## Dual timescale statistical estimation model matches human response behavior

The psychoacoustic results demonstrate that a listener's ability to detect a change in a statistical property of the environment depends on the time available to estimate this parameter, both for the pre- and post-change stimulus. However, how does the listener know, when to start estimating the new statistics? Since - as in real life - the change occurs at an unexpected time, one solution would be to compare the recent statistics to a longer term estimate of the same statistics, acting as a baseline - or 'null' - distribution. A minimal implementation of this solution consists of two processes estimating the same statistical property on different timescales (*Figure 7*).

For this purpose, we turned to models of statistical estimation of the drift diffusion type, used previously to account for visual and auditory decision making in paradigms where subjects were asked to choose between two alternative choices (*Britten et al., 1996*; *Brunton et al., 2013*). In these models a dynamic variable compares the stimulus information in favor of the two alternatives, and when reaching a predefined bound, a decision is made. We extended this model to a pair of variables, estimating the statistical property on different timescales (*Figure 7A–B* and Materials and methods). A deviation is detected if the long-term estimate (*Figure 7B*, $P_{slow}$) and the short-term estimate ($P_{fast}$) differ by more than the difference between the thresholds (*Figure 7B*, $T$). As introduced above, this was intended to capture the dual task the participants faced in our paradigm, namely to estimate the base (initial) statistics while simultaneously scanning for deviations from these statistics. The modified model is governed by four parameters, which control the timescales of the dynamics variables and the threshold. To make the model applicable to our auditory textures, we assume that multiple copies of it operate in parallel in different frequency channels (see Materials and methods).

We presented an analogous stimulus to the model, exhibiting a change in the probability of tone occurrence at a random time (*Figure 7A* left and 7B, gray, only one frequency bin shown) and in a random frequency location, and quantified the model's response in performance and response time. The model exhibited a comparable behavior on individual trials as humans (*Figure 7C*, compare to *Figure 1C*), with an initial hesitation to respond, and a mixture of false alarms (gray), correct response (black) and misses (not shown). We quantified the performance (performance, false alarms,

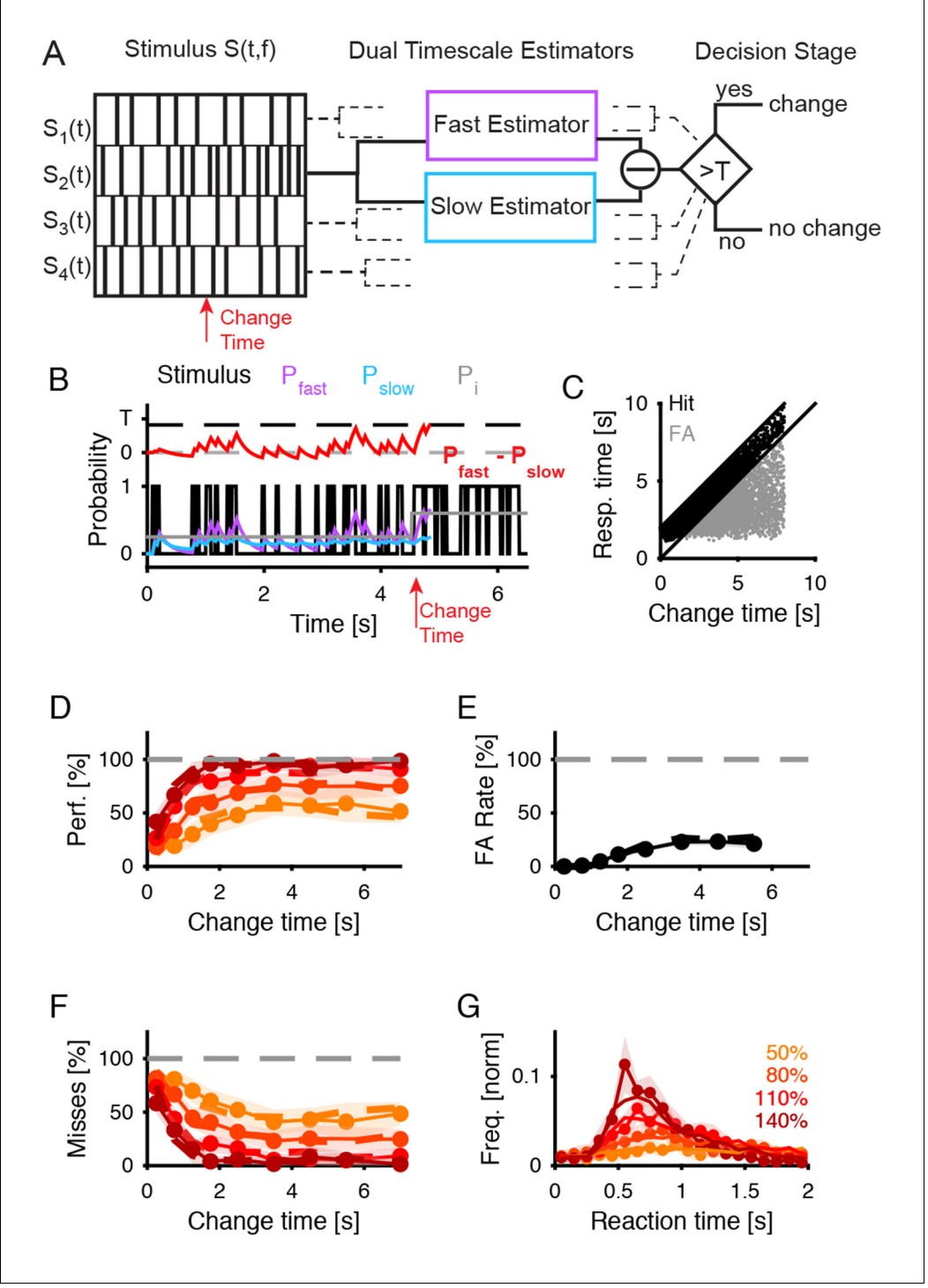

**Figure 7.** Dual timescale statistical estimation replicates behavioral results. (**A**) The dual timescale model consists of two dynamical estimation processes operating with different speeds. If their estimates differ by more than a threshold $T$, a change in the stimulus is detected. The model was fitted to the entire set of behavioral data (**D–G**). (**B**) In a single trial the slow ($P_{slow}$, blue) and the fast ($P_{fast}$, purple) estimates of the actual stimulus probability (light grey) vary with the stimulus (black) on different timescales. Here, a decision. ($P_{fast} - P_{slow} > T$) is detected at 300 ms after the change in the stimulus (red). (**C**) The distribution of response times compared with the change times exhibits a similar shape as for the real subjects (see *Figure 1B*). (**D**) Detection performance of the model (dashed lines) closely matches the human data (continuous line with 1 SEM error hull) both as a function of change time and change size (different shades see legend in **G**), see text for parameter values. (**E**) False alarm rates are also

*Figure 7 continued on next page*

*Figure 7 continued*

matched closely (same legend as in **D**). (F) Miss rates are matched equally closely (same legend as in **D**). (G) Response time distributions are also matched closely, which is of interest as no explicit model of response times was included in the model (same legend as in **D**).

___

misses) and the reaction times as a function of the change times and the change size (*Figure 7D–G*). The match between the human data and the model was close, with an average residual (mean squared error) of 0.049 (in units of probability). The correlation coefficients between the real data and the fit were [0.97,0.99,0.98], for performance (*Figure 7D*), false alarms (*Figure 7E*) and misses (*Figure 7F*), respectively.

The reaction times could be accounted for both in mean and distribution for different change sizes (r = 0.95, MSE = 0.009 (norm. prob.), *Figure 7G*). For the condition with the biggest step, a certain fraction of the responses occurred very early, which may be subject-dependent and we were unable to replicate in the present model.

The parameters that best fit the average human data were $\tau_f$ = 0.2 s, $\tau_s$ = 1.1 s, $\tau_a$ = 0.65 s, and $T$=0.40 (in units of probability). Hence, the time constants of the fast and the slow processes differed by more than fivefold, and the threshold for detecting a step was surprisingly high. The time for eliciting a motor signal was consistent with the asymptotic times we found in the human data (see *Figures 3B* and 140 %). The time constant of the transitional period represents (as the other parameters) an average over the subjects. Inspecting individual subjects revealed some variability in their propensity to react early (min median: 0.77 s; max median: 1.03 s).

The residual differences in the fit could be a consequence of the fact that the data from multiple listeners was pooled, rather than fitted individually. With the current limitation of ~1000 trials / listener, a single listener fit would be dominated by within-subject variability across trials, requiring more trials before stabilizing.

In summary, the dual timescale estimation model captures the human performance and reaction times well, suggesting that its basic principle may be implemented by the brain. The fitted timescales of estimation suggest that a rapid estimate of the present statistics can be formed within 200 ms. While this time appears sufficient to reliably distinguish the larger steps in statistics, it is insufficient to detect small changes in occurrence probability, which are often perceived as unchanged statistics.

In relation to the CPP's response properties, it is noteworthy that the decision variable in the dual-time scale model exhibits a similar, positive dependence between slope and change size (evident from Equations 2/3/3). If the evidence accumulation continued during the time interval between decision commitment and the actual motor execution, this would translate into a dependence of the height on the change-size as well. In agreement with the CPP dynamics, the slope should not depend on change time. Instead, the estimate of $P_{slow}(t)$ should exhibit better convergence as a function of change time, leading to improved discrimination against $P_{fast}(t)$. This non-decision period separating the crossing of the threshold and the actual response execution is implicitly incorporated in the model as the motor-related increment in reaction times.

## Detection of changed statistics based on spectrotemporal processing in auditory cortex

The dual timescale model successfully captures human performance via an estimation of stimulus statistics. While this suggests a consistency with the principle of statistical estimation, it does not provide any insights into putative neural implementations. For this purpose, we turn to an established model of auditory cortical processing ('cortical model', *Chi et al., 2005*; *Elhilali et al., 2009*; *Krishnan et al., 2014*; *Patil et al., 2012*; *Yang et al., 1992*), which we augment here with a decision stage specific to the present task. In particular, this alternative model investigates whether the cortical model (and hence the primary auditory cortex) represents the acoustic stimulus in a way that supports an account of our psychoacoustic data, i.e. supports decision making in certain statistical contexts.

The cortical model emulates the spectrotemporal response properties of neurons in primary auditory cortex, which have been extensively studied by various groups (*Ahrens et al., 2008*;

*Eggermont, 2002*; *Kowalski et al., 1996*). Its responses are based on a filterbank-based, joint spectrotemporal modulation analysis following the output of the early stages of the auditory system (auditory spectrogram, *Figure 8A*). Parameters and properties were set to approximate the responses of neurons in auditory cortex (see Materials and methods for details) (*Chi et al., 2005*; *Yang et al., 1992*). The spectrotemporal filters cover the experimentally observed range of 1–30 Hz and 0.5–8 cycle/oct, whose outputs are weighted in correspondence with the experimentally observed abundance of these properties in A1 (*Kowalski et al., 1996*), *Figure 8B*).

We simulated two types of readouts from the model to account for two of the main experimental constraints. For the first, we summed all cortical outputs to simulate an effective EEG recording with limited spatial separation of sources, leading to a global response. As expected, in this case, trial onsets and offsets produced strong responses (*Figure 8B*), with a plateau of sustained response for the whole duration of the stimulus. The responses due to the statistical change in the stimulus were largely diluted in the summed response and thus could not been discerned, consistent with the present EEG recordings of the auditory electrodes (*Figure 5B*).

The ranges of spectral bandwidths and timescales related to the change were kept constant over the whole duration of the task. Consequently, a more optimal strategy would be to focus on the

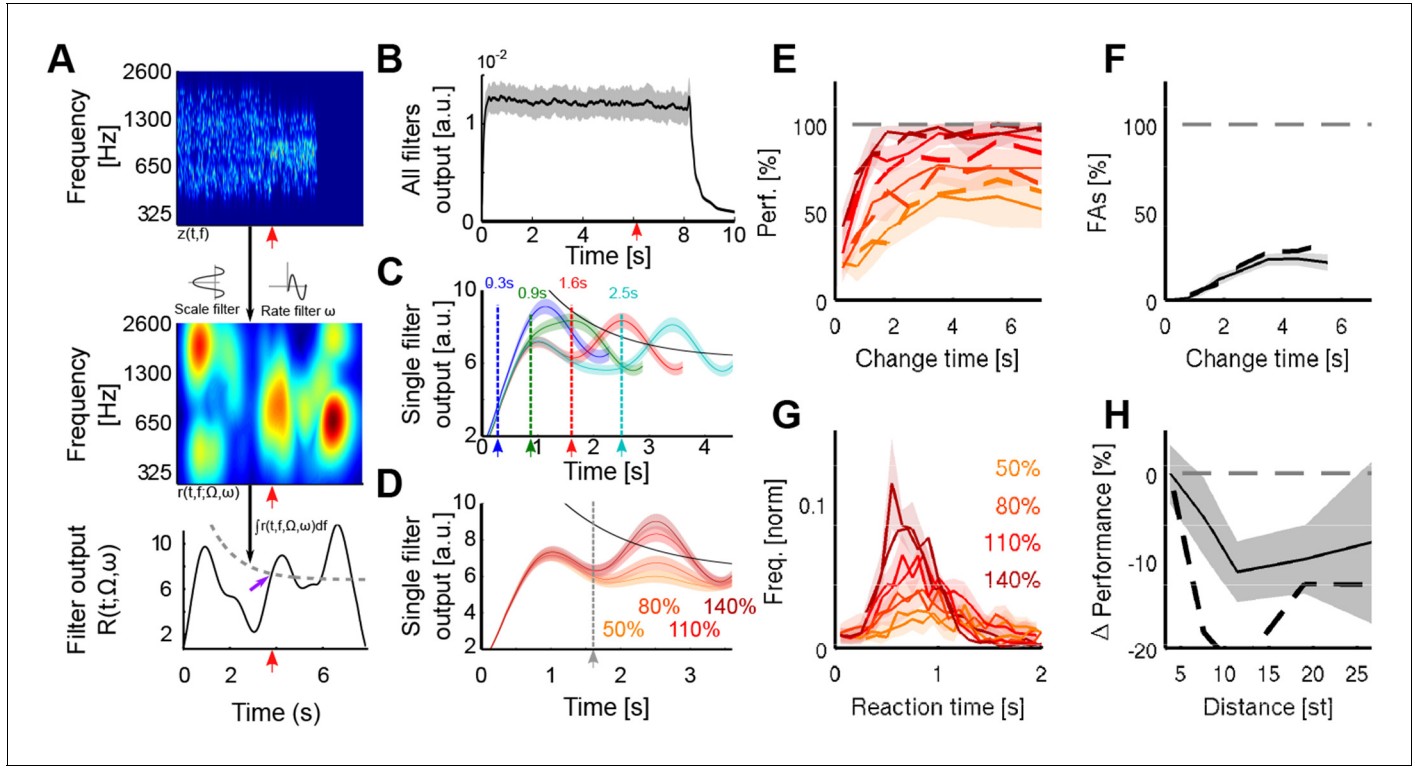

**Figure 8.** A cortical filter-bank model provides an implementation consistent with the behavioral results. (A) Conceptual structure of the model. The cochleogram (top panel) is passed through modulation filters (scale Ω: 0.54 cycle/oct.; rate ω: 0.72 Hz) for obtaining a cortical representation of the sound (middle panel). Changes are detected with a threshold (bottom panel, grey dashed line) applied to the frequency-averaged cortical representation (collapsing threshold parameters: λ = 1.14 s; b = 10.77; a = 6.23). First peak exceeding the threshold is classified as change (purple arrow). Timing of change is indicated by a red arrow in the three panels. (B) Average output of the cortical model across all modulation filters. Although trial onset elicits an overall increase in activity, the change in statistics does not lead to an average change in activity (depiction for single trial length, with change time indicated by arrow). (C) Single filter output as a function of change time (average over 100 trials for each curve). Change times are indicated by colored arrows. Notice that the change-related peak is not discernible for early changes, due to its interaction with the onset response. Same parameters than in A). (D) Single filter output as a function of change sizes (average over 100 trials for each curve). Same parameters as in A). (E) Performance for human participants (thin lines) and the decision model (dashed thick lines), as a function of change size and change time. Same colors as in D). (F) False alarm rate as a function of change size and change time. Same colors as in D). (G) Response time distributions as a function of change size. Same colors as in D). (H) Decrease in performance with respect to the distance between incremented bins. Actual data in full line, model result is depicted with a dashed black line.

temporal modulation filters in cortex that are most activated by the statistical change. Hence, we postulated that high-order areas could monitor the outputs of the task-relevant temporal filters. For example, subjects could make their decisions based on the largest output produced by the change. These would be sampled from the filter with the temporal dynamics and spectral modulation that roughly matched those of the stimuli. The response thus selected is shown in *Figure 8A*. Aside from the strong responses at stimulus onset and offset, the responses now exhibited in addition a prominent intermediate peak due to the change in statistics (*Figure 8A*). This change-induced response peak vanished in trials when the pre-change interval was very short because it became fused with the large onset peak (*Figure 8C*). Change size was encoded in the slope of this cortical response (*Figure 8D*), consistent with the neural CPP response (*Figure 5H*). The variability in responses of the cortical outputs was solely due to the random tone-clouds preceding and following the change in the stimulus.

To quantitatively simulate the perceptual decisions of the listeners, we analyzed the cortical filter outputs for individual trials. In order to take into account the time-dependence of the CPP amplitude in the EEG recordings, we used a time-varying threshold that remained identical across all conditions (see *Figure 8A* and Materials and methods). The first peak exceeding this threshold (if any) was considered to be the decision point (purple arrow in *Figure 8A*). This readout mechanism was fitted to the performance and false alarm rate across change sizes and change times by allowing five free parameters (*Figure 8E–F*): the (bandwidth) scale $\Omega$, the (temporal) rate $\omega$, and the decision parameters ($\lambda$, $a$, $b$; see Materials and methods). The parameters that best fitted the human dataset were $\Omega$ = 0.54 cycle/oct, $\omega$ = 0.72 Hz (a rate corresponding approximately to dynamics or an integration time-constant of the order of 1–2 s), and $a$ = 6.2, $b$ = 10.8, and $\lambda$ = 1.14 s ($\rho$ = 0.95; p<$5.10^{-16}$; MSE = 0.7%). The scale value corresponds to a full width at half-maximum for the scale filter of approximately 0.56 octave, very close to the frequency region spanned by localized changes (0.55 octave). This may indicate that subjects preferentially used a single scale value for monitoring the frequency modulation and that they estimated the most common frequency modulation across trials since half of the trials contained localized changes. Importantly, reaction times predicted by the model matched subject reaction times remarkably well both in distributional shape, mean and spread (*Figure 8G*), although the fitting procedure did not make use of this information ($\rho$ = 0.90; p<$5.10^{-13}$; MSE = 9.1%).

Scale filters integrate frequency modulations over a limited spectral bandwidth set by the scale factor $\Omega$. Such scale filters are more prone to detect changes localized in the spectrotemporal modulation domain. Therefore spectrally distributed changes could be missed by the decision stage as they elicit less activity in the filter outputs. This is reminiscent of the observation that listeners detected changes more efficiently if their energy was concentrated in the frequency domain (*Figure 4A*). Consistent with this, we found a decrease in the model performance for non-localized changes, without fitting the model parameters to this aspect of the data (*Figure 8H*).

Thus, the model describes a physiological mechanism that accounts for the behavioral data, as well as suggesting an implementation for the basis of statistical estimation in neural terms. In relation to the neural responses, it provides an interpretation for the lack of change-related signal in the auditory EEG electrodes, and the decision signal's slope also scales with the change size (*Figure 8D*), i.e. the amount of evidence. Similar to the dual timescale model, the peak size of the decision signal increases with change size (*Figure 8D*), unless interrupted by the threshold. The reduction of the threshold over time (to compensate for the task designs) is consistent with the experimentally observed reduction in CPP size as a function of change time (*Figure 5M*).

## Discussion

We investigated how listeners detected changes in spectrotemporally broad acoustic textures, as a model for change detection in complex auditory environments. The results demonstrated that listeners estimated the statistics of the stimulus to make their decision, as evidenced by the dependence of performance, reaction times, and the CPP response on the stimulus parameters. We developed a drift-diffusion type model for estimating certain stimulus statistics, which accounted well for the response performance and dynamics in human listeners. Finally, we adapted a model of auditory cortical processing to provide a link between statistical estimation and the underlying physiology. The model accounted equally well for the human performance by exploiting a range of temporal filters,

providing a potential, neurally plausible substrate for statistical decision-making. The decision signals of both models exhibit consistent integration behavior with the CPP potential.

## Relation to previous spectral detection tasks

The present experimental paradigm mimics the unexpected transformation of a sound source within a natural auditory environment. There are some relations to previous research on spectral representations of sound, e.g. profile analysis (*Green, 1992*; *Green and Berg, 1991*; *Hartmann, 1986*; *Lentz and Richards, 1997*; *Neff and Green, 1987*). Our work, however, differs in several relevant ways. In profile analysis, subjects detected spectral shape changes on static spectra that were presented in isolation for short times (fraction of a second each). By comparison, our stimuli were dynamic and sustained (multiple seconds), and changes were detected in the midst of a continuous background with an explicit measure of reaction times. This enabled us to explore the dynamic acquisition of the statistical information.

Further, a series of recent studies investigated detection of change occurring in first- and second-order sound statistics (*Sohoglu and Chait, 2016*; *Barascud et al., 2016*). In particular, these authors probed the detection of appearing or disappearing regular sound sources in an acoustic scene (*Sohoglu and Chait, 2016*). This type of changes featured modifications of first- and second-order sound statistics, which also included an increase in the overall sound level. In comparison, our stimulus design allowed us to limit the change to the first-order statistics while keeping the overall sound level constant.

Our experimental task offers a compromise between complexity of spectrotemporal structure *versus* tractability and interpretability of the changes. Furthermore, the task design and acoustic stimulus are well-suited for electrophysiological studies with behaving animals, where one can easily estimate neuronal receptive fields from the responses to tone clouds at the same time as the animal detects the changes (*Ahrens et al., 2008*; *Wang et al., 2012*).

Another important aspect of the experiments was their interleaved (as opposed to block-based) design for change sizes and other parameters, which had several consequences. For instance, it is likely that the observed performance underestimated optimal performance, since the time, location and size of changes were unexpected. This also prevented subjects from using a template-match strategy on the largest change size, and provided access to reaction times, which consistently mirrored performance, and perhaps the certainty of the subjects in their decisions (*Kiani et al., 2014*).

## Modelling statistical decision-making on two levels

Following the modeling steps proposed by *Marr (1982)*, we provided an *algorithmic* and an (neural) *implementational* model of our subjects' behavior. The *algorithmic* approach implemented the principle of statistical estimation, while the neural model leveraged principles of auditory cortex processing. Although both models analyzed recent inputs, and effectively detected deviations from them, they differed fundamentally in their levels of description and abstraction.

The statistical estimation model implements the principle of statistical integration in a close-to minimal form, and provides a link to classical drift-diffusion models. It is a mechanistic, non-neural description of the process that performed statistical estimation in the classical sense, by representing and comparing the probability of stimuli in frequency bins, based on a lossy memory. Previous work has suggested a possible neural implementation of such a decision making process, in the form of competing neuronal populations, each corresponding to one alternative choice (reviewed in *Insabato et al., 2014*). While this approach can in principle be extended to the estimation of other properties of a stimulus distribution, i.e. moments or correlations, it has to be adapted more specifically to each particular task. In the present case we chose a static set of parameters, since the change time distribution remained unchanged in a session. More generally, (temporal) integration properties can adapt to the recent statistics, as recently shown in related contexts (*Raviv et al., 2012*; *Ossmy et al., 2013*).

The cortical model differs fundamentally in that it seeks to capture basic sensory neural responses and is inspired by physiological mechanisms. In this sense, it is agnostic to the type of stimulus, and can be readily extended to handle more complex scenarios such as changes in natural stimuli, speech and music. To create behavioral performance from its representation, we merely added a filter selection and a decision criterion. The spectrotemporal filters implemented in the cortical model

exhibit alternating excitatory (positive) and inhibitory (negative) fields (*Figure 8A*) that compare the spectral stimulus properties over a given time window set by a filter's temporal rate. As such, it effectively integrates the recent input with opposing signs to detect a change, which can be compared to the difference between the fast and slow estimators in the statistical estimation model.

Therefore, we may view this model as approximating a neural implementation of the statistical model, and thus as a bridge to a neural interpretation of the behavioral performance and the EEG recordings. Several properties of human performance and of the neural data can be considered within each model's framework. The most relevant are (i) reduced performance in detecting early changes, (ii) longer reaction times for early changes, and (iii) reduced height of the centro-parietal EEG responses for late changes.

In the cortical model, the reduced performance results primarily from the large onset response masking the responses to the smaller subsequent change, rendering the peak response less detectable (i). In order to simulate the instructions to the subjects *not* to report the stimulus onset as a change, the detection threshold was set to decrease from a larger initial value, which will delay responses for early changes (ii). Interestingly, this choice for the threshold is in line with the reducing CPP potential heights as a function of change times (iii). Overall, the integration time-constants in the cortical model on the order of 1 s (due to bandpass filters tuned at rates near ~1 Hz) appear sufficiently long to explain the decision dynamics exhibited by the subjects (*Figure 2*). These time-constants, while on the slow-end of the range, are still found in the primary and secondary auditory cortical regions (*Kowalski et al., 1996*; *Liang et al., 2002*).

In the statistical model, the reduced performance (i) is a consequence of the model's design having two estimators: one with a fast and the other with an adaptive time-constant ($\tau_f$ and $\tau_s$). At stimulus (trial) onset, the absence of prior evidence is reflected by the equality of the two time-constants. As the trial progresses, $\tau_s$ becomes longer, and the difference between the two estimator outputs increases to reflect the buildup of evidence for a change in stimulus statistics (see Materials and methods). the dynamics are a consequence of the time-constant dynamics (as above) as well as the not-yet converged estimate of the initial occurrence probability (ii). There is no correspondence for the observed decrease of the CPP potential as a function of change time (iii).

It has previously been proposed that subjects may be trading response speed for accuracy (*Teichert et al., 2014*). We think this may apply to the first period up to 1 s, where subjects responded very little. After this point accuracy quickly rose (*Figure 2C*), as did the false alarm rate (before reaching its plateau). The time controlling the divergence between the two estimators in the statistical model ($\tau_a$ = 0.65 s) roughly matches this time scale and may be accounting for an initial postponing of decision by the subjects. An alternative modelling strategy would include a dead-time, corresponding to the minimal time subjects take before responding. While this appeared unnecessary for the present data, such a model may become relevant if the paradigm includes blocks of different response window length, where subjects are forced to respond more quickly to perform successfully.

In summary, what is typically termed accumulation of evidence (and its associated performance and dynamics) could be explained by the dynamics of the onset response in the cortical model intertwined with its integration time-constants. Future experiments need to further test the validity of this neural interpretation, given the ubiquity of such 'sudden' events in natural stimuli due to saccades (in vision), attentional switches, or trial onsets, which could also influence the detectability of changes (as e.g. in change blindness, *Levin and Simons, 1997*; *Rensink et al., 2000*).

## EEG recordings and the site of decision-making

As discussed above, recognizing a change in the statistics of a complex spectrotemporal sound requires the extraction and accumulation of evidence from the stimulus to estimate decision-relevant properties. This transition from a stimulus-related to a task-related representation needs to occur along several stations of the auditory system. Our EEG recordings provide partial evidence regarding their putative location. Specifically, we found a clear difference in the representation of the stimulus at the central electrodes (estimated to originate from auditory cortex activity) and at the centro-parietal electrodes (estimated to reflect parietal activity): while the central electrodes exhibited a sharp onset response at stimulus onset (*Figure 5A*) and offset, they showed little evidence of the change response or of the presumed accumulation of evidence for a change (*Figure 5B*).

In sharp contrast, the centro-parietal electrodes displayed no response to the onset (*Figure 5D*), and a clear evidence of the sensory evidence accumulation after the change aligned to response (*Figure 5E2*). Previous studies using a linear increase of sensory evidence found a quadratic time progression of the centro-parietal potential. In the present task, the constant amount of evidence as function of time resulted in more linear dynamics of the centro-parietal potential, supporting the integration hypothesis. The task-irrelevant abrupt sensory event (i.e. the onset) was thus filtered out in the parietal EEG response while the task-relevant event (the change in statistics), although more subtle in nature, was selectively integrated and converted into a decision signal.

A set of related EEG studies termed the corresponding potential the centro-parietal positivity (CPP, *Kelly and O'Connell, 2013*; *O'Connell et al., 2012*). Consistently, we found the CPP slope and amplitude at the response time to be correlated with evidence level (change size). However, the amplitude was independent of reaction time, and did not depend on change size before the response time (200–100 ms). Additionally, the amplitude of the CPP was found to be inversely related with change times, as one would expect if performance would be influenced by the estimated, maximal trial duration. Such a reduction would be expected both as a consequence of the general task design, i.e. on the one hand, the requirement to not respond to the onset of the first texture, and on the other hand, the possibility to approximate the maximal trial duration. For practical reasons, arbitrary trial lengths are not realizable. Hence, subjects could form an expectation of the maximal sound duration, which means participants could be subject to an urgency signal that would lower their criterion in this time-range.

On the other hand, the texture onset may also play an important role, since the requirement to not respond to an otherwise salient change, may be regulated by a change in threshold (as modeled in the decision stage of the cortical model).

A decrease in decision threshold has been used in models of decision-making for dealing with speed-accuracy trade-offs (*Bogacz et al., 2006*) and observed in electrophysiological studies (*Heitz and Schall, 2012*). Although it could in theory explain the time-dependence we observed for CPP heights, subjects did not exhibit any urgency to respond, even after a few seconds, as exemplified by the constant FA rate per unit of time (*Figure 2D*). Instead, we propose that this decrease in threshold reflects a more sensitive criterion for change detection, via a more settled estimate of the initial statistics. The decision threshold would thus be dynamically adjusted during the course of a trial. Importantly, the lack of increase in FA rate suggests that the improved estimate of the initial statistics would also reduce the neural response to expectable deviations, such that the sensitivity (type I errors) stays at the same level.

This predominance of decision-related signals in the centro-parietal electrodes is consistent with decades of research in the accumulation of task-related visual information in the parietal cortex, more specifically in decision-making with saccades in the lateral intraparietal (LIP) cortex (*Huk and Shadlen, 2005*; *Roitman and Shadlen, 2002*; *Shadlen and Newsome, 2001*). Neurons in LIP have been shown to exhibit activity correlated with the accumulation of visual evidence coming from MT (*Huk and Shadlen, 2005*; *Mazurek et al., 2003*). Their firing rate usually exhibits a linear increase until the animal makes a decision (*Shadlen and Newsome, 2001*). In these studies, typically a fixed threshold on neural firing rate is used to relate neural activity to decision making.

It has recently been suggested that individual neurons change their firing rate instantaneously at the single trial level (*Latimer et al., 2015*). We presently observed gradual, rather than step-wise changes in our across-trial averages. However, we predict that even single trial EEG signals would be gradual as these step-changes occur randomly between neurons, and hence are unlikely to be synchronized at the population-level. Due to the large ensemble of neural responses contributing to a single scalp location's potential, this instead results in the commonly seen ramping activity on the EEG level, as observed in our data.

The lack of evidence for a change-related signal in the auditory EEG potentials can, however, not fully rule out the presence of a change-related signal in auditory cortex in the present stimulus context. The representation of the change could be diverse and distributed, which may average out in the non-selective, coarse averaging on the EEG level (see *Figure 8B*). This is also consistent with recent work, demonstrating choice-related signal in auditory cortex (e.g. *Bizley et al., 2013*; *Tsunada et al., 2016*). Our cortical modelling suggests that the representation in auditory cortex provides a good substrate for initial accumulation of sensory information about changes in stimulus statistics, which is then selected and amplified in parietal cortex, leading up to the sustained parietal

activity and a full representation of accumulated evidence and choice (**Shadlen and Newsome, 2001**). This interpretation is supported by the match in performance, reaction times (**Figure 8E–G**), and in the progression of activity between specific filters of the cortical model (**Figure 8C**) and neural data (**Figure 5E**).

In conclusion, as with many other cognitive functions, it is likely that higher-order areas such as the LIP and PFC select and potentially amplify task-relevant outputs of the auditory cortex. To test this hypothesis and the value of the proposed models, it will be necessary to extend change detection tasks to more natural and complex stimuli. As shown previously (**Lewicki, 2002**; **Smith and Lewicki, 2006**), natural statistics shape neural processing, and in a similar way should be informative about which changes to focus on in research. Furthermore, the models should be extended to include the effects of cognitive functions in modulating this process, such as attention or expectations.

## Materials and methods

### Participants

In the main psychophysical study, 15 normal hearing subjects (mean age: 24.8y, 6 females) participated, 10 of which could be included for final analysis (see below for criteria). A different set of 18 subjects participated in the combined psychophysics and EEG experiment (mean age: 30 ± 10 years, 7 females), all of which could be included for final analysis (see below for criteria). All experiments were performed in accordance with the guidelines of the Helsinki Declaration. The Ethics Committees for Health Sciences at Université Paris Descartes approved the experimental procedures.

### Experimental setup

*Acoustic Stimulation* Subjects were seated in front of a screen with access to a response box in an acoustically-sealed booth (Industrial Acoustics Company GmbH). Acoustic stimulus presentation and behavioral control were performed using custom-written software in MATLAB (BAPHY, from the Neural Systems Laboratory, University of Maryland, College Park; available upon request). The acoustic stimulus was sampled at 100 kHz, and converted to an analog signal using an IO board (National Instruments, PCIe-6353) before being sent to diotic presentation using high-fidelity headphones (Sennheiser i380, calibrated flat, i.e. ±5 dB, within 100–20000 Hz). Reaction times were measured via a custom-built response box and collected by the same IO card sampled at 1 kHz.

*Electroencephalogram (EEG) acquisition* EEG recordings were collected in a separate set of 18 normal-hearing subjects while listening and responding to the texture change stimuli. Continuous EEG data were recorded using a 64-channel system (ActiCap, BrainProducts, Gilching, Germany) at a sampling rate of 500 Hz with one reference and one ground electrode. In order to standardize electrode placement on the skull, we used a default fabric head-cap that holds the electrodes (EasyCap, Equidistant layout, 60 scalp, four ocular channels). The analysis of EEG responses was carried out offline (see section Data analysis).

### Stimulus design and trial procedure

We investigated the conditions under which listeners could detect a change in the statistics of complex acoustic stimuli. More specifically, we wondered how subjects capture the percept of a spectro-temporally complex stimulus, and then use it as a background to detect changes relative to it. Concretely, in an experimental trial, a sound texture was presented, allowing the subjects a randomly varying period of time to form a percept of the stimulus (i.e. 'estimate the baseline statistics'), and then a change in the frequency distribution of the tones was introduced (while maintaining the overall sound level). After the change, subjects had up to 2 s to indicate that they had detected it. The stimulus captures the central textural properties of complex spectrotemporal structure and statistical predictability. Both the stimulus design and the procedure are described in detail below.

#### Stimulus design

Briefly, the stimulus was a 'cloud' of tones, i.e. a train of short pure tones chords (30 ms) drawn from a range of 2.2 octaves (400 to 1840 Hz), where the occurrence probability of each tone was governed by a marginal distribution (see below, **Figure 1**, and sound examples in **Supplementary files**

*1–4*). The frequency resolution of the tone distribution was 12 semitones per octave, i.e. 26 logarithmically spaced pure tones covered the used frequency range. To limit the number of experimental conditions, these were grouped into eight spectral bins, each comprising 3–4 of the pure tone frequencies (see *Figure 1* for illustration). The marginal distribution was chosen to ensure that the actual rate of tones per bin was controlled, independent of the number of pure tone frequencies constituting the bin. The entire stimulus can be described by a spectrogram denoted by S(t,f) as a function of time and frequency.

The minimal temporal unit of the stimulus was a 30 ms chord, i.e. a synchronous presentation of multiple pure tones. The number of tones for each chord was drawn from a Poisson distribution with a fixed mean of 2 tones per octave. The mean number of tones per chord was kept fixed as a function of time to avoid changes in level (see below). The frequency of each tone in a chord was chosen in two successive steps: First, one of the eight spectral bins was selected according to a marginal probability distribution (see below). Second, within this bin, one of the pure tone frequencies constituting the bin was randomly selected. Chords at different times were drawn independently from each other.

The baseline marginal probability distribution was composed of 8 frequency bins with discrete probability values (*Figure 1A*, left). These values were chosen pseudo-randomly for each trial, forcing subjects to always reestimate the stimulus statistics. The probability in each bin could take one of 3 values: 0.083, 0.125, 0.188. To avoid differences in spectral density, the number of bins with each probabilities was fixed to three bins with p=0.083, two bins with p=0.125 and 3 bins with p=0.188. The marginal distribution is thus normalized, i.e. the sum across bins equals 1. Since multiple pure tone frequencies constituted each of the eight bins, the probability per pure tone frequency bin was correspondingly lower: based on this marginal distribution and the number of tones per chord, the effective probability of a tone falling in a pure tone frequency bin thus ranges between 0.021–0.063 per chord duration, corresponding to an average rate of ~147 tones/s.

The change in statistics consisted in a change in the baseline marginal distribution. Two out of the eight spectral bins were increased in probability at a random point in time (referred to as *change time*, more details below) during stimulus presentation, i.e. the stimulus continued uninterrupted. The increment size will be referred to as *change size* and was drawn from a set of discrete values: 30, 50, 80, 110, 140% (inset in *Figure 2A*), relative to the single bin probability in a uniform distribution (for eight bins this is 0.125, i.e. a 50% change size would be an increment of 0.0625). In order to exclude cues from global level changes, the marginal distribution was simultaneously renormalized, thus keeping the global level constant within a trial (i.e. as mentioned above the rate of tones per chord was kept constant at all times). Since the 30% condition was only collected for three subjects, it is omitted from most plots, although results were generally consistent with the other conditions.

The relative *spectral locations* of the two changed bins were separated into two conditions:

1. *Localized*: the frequency bins containing the change were adjacent. To limit the number of conditions, only 4 pairs of bins, {1,2}, {3,4}, {5,6}, {7,8} were tested at all increment levels.
2. *Non-localized*: the frequency bins containing the change were separated in frequency. To limit the number of conditions, we chose a subset of distances (D=[2, 3, 5, 7] bins, i.e. [6.6, 9.9, 16.5, 23.1] semitones (st)) and only used the change size 110% (determined as intermediate difficulty during pilot studies). Since certain inter-bin distances are more frequent (i.e. six for D = 2, five for D = 3, three for D = 5, one for D = 7), the number of trials going into each distance differs, which scales the error bars accordingly (see *Figure 4B*).

The time at which the change occurred (*change time*) was drawn randomly from an exponential distribution (mean: 3.2 s) limited to the interval of [0,8] s (*Figure 1B*). This choice of distribution prevents subjects from developing a timing strategy, by keeping the probability of a change constant over time. The associated flat hazard rate minimizes the ability of participants to anticipate the trial end (*Janssen and Shadlen, 2005*; *Kiani et al., 2008*). The change time is an important parameter with respect to the estimation of the first marginal distribution, with the hypothesis that greater change times improve detection of changes.

Given the low, per-bin probabilities (see above), individual tones remained distinguishable. Hence, the spectrotemporal density was low enough to avoid fusion into a single stream per channel, although the present study still has some parallels with previous paradigms, e.g. spectral shape analysis (*Green, 1988*, *1992*; *Green and Berg, 1991*) (see Discussion).

## Procedure

The experiment proceeded along three phases: instruction, training, and main experiment. After reading the instructions, subjects went through 10 min (60 trials) of training, where they were required to obtain at least a detection performance of 40%. The training comprised only stimuli of the two largest change sizes (110%, 140%). Three subjects in the psychophysics-only group did not attain the criterion level of performance and were not tested further.

The main experiment was composed of two sessions of about 70 min each, comprising a total of 930 trials. The two sessions were never more than two days apart. Each session contained three blocks of about 20 min, for a total of 465 trials per session, corresponding to 30 repetitions of each condition (for the three subjects in which the 30% condition was tested the total trial number increased to 1050). In between blocks subjects could take a short break.

The instructions specified that subjects would be compensated according to their performance, although an easily attainable threshold of proficiency would give them the maximal compensation. However, all subjects were compensated equally according to the length of the experiment (€10/hour).

After reading the instructions, subjects were aware that the change could arise at any moment on each trial and that their task was to detect it within a 2 s window. When subjects heard a change, they pressed a response button. This terminated the trial and the sound. Hence, the subject had up to 2 s after the change to detect the change in stimulus statistics.

Visual feedback was always displayed on a screen in front of them after the end of the trial. A red square was displayed, if the button was pressed before the change (false alarm), or if the button was not pressed within the 2 s time window after the change (miss). A green square was displayed, if the button was pressed after the change, but within the 2 s window (hit).

In addition, stimulus level was roved from trial to trial, chosen randomly between 60 and 80 dB SPL. This procedure is classically applied to prevent subjects from adopting an absolute level strategy (*Green and Berg, 1991*). Overall level was not found to significantly influence performance (p=0.89, ANOVA). The inter-trial interval was ~1 s with a small, random jitter (<0.1 s) depending on computer load.

## Psychophysics procedure during EEG recordings

During the EEG recordings, stimuli and experimental procedures identical to those of the psychophysics experiments were used. In addition, subjects were required to continuously fixate a white cross on the screen. They were asked not to blink and to keep fixation especially during the sound presentation. After the end of the trial (i.e. either the end of the sound or their response), they received a visual text feedback after 0.5 s. After the feedback disappeared, eye blinks were allowed during the intertrial interval indicated by on-screen text underneath the fixation cross. At 1 s before the next stimulus, the text disappeared, indicating that blinking should be prevented subsequently.

## Data analysis

The ability of the subjects to detect the change in stimulus statistics was quantified using two measures, performance and d-prime, denoted d'. These analysis (*Figures 1–4*) were performed on the data obtained during the psychophysics experiments, and restricted to the trials embedding localized changes unless stated in the text. In addition, reaction times dependences over stimulus parameters were analyzed.

## Performance

We computed a subject's performance as the fraction between successful detection (hits) out of the total trials for which the change occurred before the response (hits + misses). False alarms were excluded from the performance computation, since the responses occurred before the change arose (see d' for an inclusion of false alarms).

## d' analysis

We developed a time-dependent d' measure, in which longer trials serve as catch trials before the change occurs (*Green and Swets, 1966*). We computed d' values to assess the ability to detect changes (*Egan et al., 1961*), while taking their false alarm rate into account, as classically analyzed

using signal detection theory. Due to the present task structure, d' could be computed as a continuous function of time from stimulus onset. We used the usual approximation d'(t) = Z(HR(t)) - Z(FAR(t)), where Z(p) is the inverse of the Gaussian cumulative distribution function (CDF). HR(t) is the hit rate as a function of time since stimulus onset. HR was computed as the fraction of correct change detections, in relation to the number of trials with changes occurring at t (*Macmillan and Creelman, 1991*). As detailed above, a correct detection had to occur within 2 s of the change time. Similarly, the false alarm rate FAR(t) was computed as the number of false alarms that occurred over all 2 s windows (starting at t), in which no change in statistics occurred. This created an artificial reaction time for each false alarm, that we used for comparing the distributions of the actual reaction times resulting from the Hits (*Yin et al., 2010*). d' was computed for different times and change sizes, yielding only a limited number of trials per condition. To avoid degenerate cases (i.e. d' would be infinite for perfect scores), the analysis was not performed separately by subject, but over the pooled data. Confidence bounds (95%) were then estimated by bootstrapping within the dataset. The analysis was verified on surrogate data from a random responder (binomial with p=0.01 per time bin at 40 Hz sampling rate), providing d' close to 0 as expected.

## Reaction times

We obtained reaction times by subtracting the change time from the response time in each hit trial. For each condition, the distribution of reaction times was assembled and the median reaction time computed. Note that very early and late reaction times will in some cases not correspond to actual reaction to the change in statistics, but are coincidental, which cannot be distinguished on a trial-by-trial level. The results presented for the effect of change size on performance and reaction time were computed using only the data with change in contiguous bins (localized change). Results for the trials with *non-localized* bins (at 110% change size) were qualitatively the same, however, they were excluded from this analysis to keep the number of trials per condition equal.

These measures were computed as a function of change size and change time. While change times were drawn without binning from an exponential distribution for the experiment, they were binned for analysis using bins of exponentially increasing width (in order to achieve comparable numbers of trials in each bin).

## Performance dynamics

In order to compare the performance dynamics for different change sizes, we fitted an adapted version of the Erlang CDF to the data according to:

$$P(\Delta_c, t_c) = P_0(\Delta_c) + P_{max}(\Delta_c) * \gamma(k, tc/\tau(\Delta c))/(k-1)! \tag{1}$$

where $t_c$ is change time, $\Delta_c$ change size, $\gamma$ the incomplete gamma function, $\tau$ the function rate, and $k$ controls the function shape. $k$ was kept constant across subjects and change sizes, assuming the shape of the performance curves is invariant. Optimizations were performed using nonlinear least-squares minimization on the residuals of the fit (via 'lsqnonlin' in Matlab).

To control for inattentive subjects, we set a 30% threshold for the total false alarm rate. Two subjects were discarded according to this criterion leaving a total of 10 subjects for the data analysis, with false alarm rates below 25%.

## Analysis of EEG recordings

We analyzed two signals based on the EEG: the classical auditory event-related potential (ERP), and the centro-parietal positive potential (CPP). First, slow trends were removed from all electrodes using a low-dimensional polynomial fit ('nt_detrend', from the NoiseTools Matlab toolbox by *de Cheveigné and Parra, 2014*). We verified that a classical high-pass filter (Matlab: *filtfilt*, 0.1 Hz, 15th order, 50 dB attenuation in the stop band) gave very similar results. Electrodes were low-pass filtered below 30 Hz with a 45th order Chebyshev filter using the 'filtfilt' function in MATLAB to avoid phase distortion. All electrodes were referenced to the common average potential. All trials with at least one scalp channel exceeding 500 μV at any time after referencing were discarded as artifacts. All subjects had a low or moderate rate of blinks and eye movements and could thus be included for a total of 18 subjects.

Classical auditory ERPs were estimated from the central electrode (El. one in the equidistant lay-out of EasyCap, equal to Cz; corresponding to the center in *Nie et al., 2014*). The CPP signal was based on a set of centro-parietal electrodes (El. 14, 27, 28, similar to *Twomey et al., 2015*). Trials were then extracted in the period encompassing 0.5 s before and 3 s after the sound presentation and either time-shifted to the onset or their corresponding change times (see *Figure 5*). EEG data were segmented into shorter epochs locked on stimulus onset or response time for display. The epochs were baseline-corrected relative to a 150 ms interval prior to onset, and a 200 ms interval before change time for alignments to both change and response time.

CPP amplitude was computed as the peak amplitude at the response time in a window of ±80 ms. CPP slope was the average slope in a window of 300–50 ms before response time, computed as the mean derivative of the CPP. Topographic distributions of the EEG signal were plotted with EEGLAB ('topoplot' function) (*Delorme and Makeig, 2004*).

## Dual timescale model

We assume that subjects continuously estimate a wide range of statistical properties of the acoustic environment, and are able to detect unexpected deviations in these properties for the purpose of detecting changes in the ongoing sound. Among these properties are the probabilities of having a tone in the different frequency channels. Since these are the only determining properties in our stimulus design, we developed a phenomenological model, which estimates and detects changes in the marginal tone probabilities across multiple frequency channels (see next section for a more biologically motivated model based on a cortical filter-bank).

The model consists of change-detector modules, which operate independently on a limited spectral range and whose output is combined to enable change-detection on a full spectrum. For simplicity, the spectral division of the modules was matched to the presently relevant division of the psychophysical stimulus $S(t,f)$ (see above), i.e. we here consider four modules, one for each pair of frequency bins, whose marginal probability could change. Since the modules operate independently, frequency separation is not relevant in the present model (but see below in the cortical model). For the present model, these frequency bins are referred to as $S_i(t)$ (with $i \in [1,4]$), which contain a random set of tones, adhering to the same marginal probabilities as the psychophysical stimulus.

For each frequency bin $S_i(t)$, a pair of dynamical processes $\{P_{slow}(t), P_{fast}(t)\}_i$, acts as a change detector. $P_{slow,i}$ estimates the long-term probability of the presence of a tone at a given time in $S_i(t)$, and $P_{fast,i}$ estimates the more recent probability of the presence of a tone in $S_i(t)$. The dynamics of the processes are given by:

$$\begin{aligned} \frac{dP_{fast,i}(t)}{dt} &= \frac{P_{fast,i}(t) - S_i(t)}{\tau_f} \\ \frac{dP_{slow,i}(t)}{dt} &= \frac{P_{slow,i}(t) - S_i(t)}{\tau_s} \end{aligned} \qquad (2)$$

where $\tau_s > \tau_f$, which separates the speed of the processes. Normally, $P_{fast,i}$ and $P_{slow,i}$ are going to have similar values, since $P_{fast,i}$ is simply tracking faster than $P_{slow,i}$. However, if a change in the probability of occurrence occurs in the stimulus, the difference between the two processes will grow, since $P_{fast,i}$ will react faster to this change. A change in the environmental statistics is hence detected, if $|P_{fast,i} - P_{slow,i}| > T$, where $T$ is a threshold and a free variable of the model. Identical models exist for different frequency channels $S_i(t)$. If $T$ is exceeded in a particular $S_i(t)$, this is considered as a detected change in the environment at the corresponding time $T_i$. Hence, only the first detected change in any $S_i$ is recorded as the response. The time of actual response is then given by $T = T_i + T_m$, where $T_m$ is a constant time equals to 250 ms to account for the non-integration related process, such as stimulus representation and motor execution, up to the button press (akin to the non-decision time, by *Ratcliff and McKoon [2008]*). The model is termed a *dual timescale model*.

If we use the model as described so far, it would - correctly - detect a change in statistics at the onset of the stimulus (transition from silence to stimulus). In the present task design, the subjects were instructed to ignore the change associated with the start of the stimulus, but only detect the change in statistics within the stimulus. As laid out in the introduction, two estimations needed to be performed simultaneously: (1) estimate the probability from stimulus onset, (2) compare this estimate to the changed probability in the latter part of the stimulus (which occurs at an unknown time). To account for this initial period of estimation, we change the dynamics of $P_{slow}$ (the slower tracking

process) as a function of stimulus time. Intuitively this means that $P_{slow}$ and $P_{fast}$ initially operate on the same timescales, and thus θ is never exceeded. The modified equations therefore become

$$\frac{dP_{fast,i}(t)}{dt} = \frac{P_{fast,i}(t) - S_i(t)}{\tau_f}$$

$$\frac{dP_{slow,i}(t)}{dt} = \frac{P_{slow,i}(t) - S_i(t)}{\theta(t)} \tag{3}$$

$$\theta(t) = -(\tau_s - \tau_f)e^{-t/\tau_a}$$

The speed at which the tracking dynamics diverge is regulated by $\tau_a$. Overall, the model has four free parameters ($T$, $\tau_f$, $\tau_s$, $\tau_a$), which were matched to account for the experimentally collected data. The phenomenological model accounted for the dependence of performance on change time and change size. Given the numerators in (2) and (3), the slope of the both $P_{slow}$ and $P_{fast}$ and their difference will depend on change size (compare to the EEG data in *Figure 5*). Simulations were run at a sampling rate of 100 Hz. Fitting was performed by exhaustive search in the parameter space to avoid local minima and biasing by initial values.

The model structure is inspired by earlier accounts for decision-making in random-dot motion stimuli, i.e. so-called drift-diffusion models (*Bogacz et al., 2006*; *Britten et al., 1996*), which have also recently been used to account for acoustic click-rate comparison tasks (*Brunton et al., 2013*). In contrast to these models, the dynamical process $P_{slow}$ in our case becomes an estimate of the medium-term occurrence probability, and $P_{fast}$ an estimate of the recent occurrence probability, and a decision is made across the set of estimators (similar to *Churchland et al., 2008*) Note, that the processes can transiently exceed 1, however, on average the right hand side of the dynamical equations is zero, when the dynamical process equals the probability that Si is drawn from.

## Auditory multiresolution cortical model

The cortical model is an approximation to the analysis performed up to primary auditory cortex, which has been used successfully in a range of different auditory projects. A full description of the model can be found in *Chi et al. (2005)* and *Yang et al. (1992)*, but an outline of its basic principles is provided below.

### Computational structure of the cortical model

The cortical model processes the audio signal via two stages, inspired by the auditory pathway up to the midbrain and by the primary auditory cortex. The first stage transforms the sound into an auditory spectrogram, and the second performs a spectrotemporal analysis on this spectrogram.

The processing of the acoustic signal in the cochlea is modelled as a bank of 128 constant-Q, asymmetric bandpass filters, equally spaced on the logarithmic frequency scale spanning 5.3 octaves. The cochlear output is then transduced into inner hair cell potentials via a high-pass and low-pass operation. The resulting auditory nerve signals undergo further spectral sharpening via a lateral inhibitory network. Finally, a midbrain model resulting in additional loss in phase locking is performed using short term integration with a time constant of 4 ms, resulting in a time-frequency representation (the auditory spectrogram z(t,f)) (top panel in *Figure 8A*). The central stage further analyzes the spectrotemporal content of the auditory spectrogram using a bank of modulation-selective filters centered at each frequency along the tonotopic axis, mimicking neurophysiological receptive fields. This step corresponds to a 2D affine wavelet transform, with a spectrotemporal mother wavelet, defined as a Gabor-shape in frequency and exponential in time. Each filter h is tuned ($Q = 1$) to a specific rate ($\omega$ in Hz) of temporal modulations and a specific scale of spectral modulations ($\Omega$ in cycles/octave), and a bidirectional orientation (+ for upward and - for downward). The response of each cortical filter in the model is given by

$$r_\pm(t, f; \omega, \Omega; \theta, \Phi) = z(t, f)_{t,f}^* \, h_\pm(t, f; \omega, \Omega; \theta, \Phi) \tag{4}$$

where $*_{t,f}$ denotes convolution in time and frequency, where θ and Φ are the characteristic phases of the cortical filter, which determine the degree of asymmetry in the time and frequency axes respectively (middle panel in *Figure 8A*). Because changes were isotropic within the sound spectrum, we averaged the upward and downward components of the scale modulation filter. To simplify the

analysis, we limited our computations to the real cortical outputs across frequency (i.e. responses corresponding to zero-phase filters). The resulting modulation response is denoted R(t;ω,Ω) (bottom panel in *Figure 8A*). Simulations were run at a sampling rate of 100 Hz.

### Decision process based on the cortical model output

On a single trial basis, the stochastic nature of the stimulus was reflected in the noisy outputs of the cortical model. To facilitate change detection on single trials, we post-filtered the modulation response R(t;ω,Ω) using the average response to a change in statistics. Concretely, the shape of the trial-averaged response in R(t;ω,Ω) was convolved with single trials, to improve detection of change. Due to the different modulation rates, the length of the average response shape varied by modulation rate $\omega$ as 1/ (2$\omega$) ms. A unique combination of rate ω and scale Ω was used across all trials to characterize the modulation response. Next, we implemented a decision criterion on top of the filtered R(t;ω,Ω).

Due to the comparative nature of the present paradigm and because the onset peak was not driven by any task-relevant feature of the sound, a time-dependent decision boundary was better suited to match the experimentally observed reaction times in both models. This was inspired by previous studies that described either time-varying collapsing boundaries (*Ditterich, 2006*) or linearly increasing emergency-related gain (*Cisek et al., 2009*; *Drugowitsch et al., 2012*). We designed the time-dependent threshold as follows:

$$\mathrm{T(t)} = \mathrm{be}^{-\mathrm{t}/\lambda} + \mathrm{a}$$

(5)

where *a* and *b* scales the amplitude of the threshold and λ sets its time-dependence. The first peak exceeding the time-dependent threshold was labelled as the decision timing.

In total, the decision stage is controlled by five parameters: the time-varying threshold (λ, *a*, *b*), the scale Ω, and the rate ω, while other parameters of the cortical model were kept fixed. The threshold parameters tune the balance between conservative and liberal decisions. To take into account this aspect we fitted both performance and false alarm rate across all subjects for all change sizes and change times. Motor-related delay was accounted for by a 250 ms offset added to the estimated reaction times, as was done for the phenomenological model.

## Statistical analysis

If not specified otherwise, nonparametric tests were used. When data were normally distributed (for performance), we checked that statistical conclusions were the same. One-way analysis of variance was computed with the Kruskal-Wallis' test; two-way using Friedman's test. Error bars are ±2 SEM (standard error of the mean), unless specified otherwise. All statistical analysis was performed using Matlab (The Mathworks, Natick).

## Acknowledgements

We would like to thank the Equipe Audition, the Donders Center for Cognitive Neuroscience, and the NSF-funded Neuromorphic Cognition Engineering Workshop in Telluride, CO, USA, for allowing us to use sound booths and EEG equipment, with helpful discussions. Funding was provided through the ERC ADAM project. ANR-10-LABX-0087 IEC and ANR-10-IDEX-0001–02 PSL* supported the research unit. SS was also supported by an ARO grant 63113-LS. BE was supported by a European Commission's Marie Curie Grant (660328).

## Additional information

### Funding

| Funder | Grant reference number | Author |
| --- | --- | --- |
| Agence Nationale de la Recherche | ANR-10-LABX-0087 IEC | Yves Boubenec<br>Jennifer Lawlor<br>Shihab Shamma<br>Bernhard Englitz |

| Agence Nationale de la Recherche | ANR-10-IDEX-0001-02 PSL* | Yves Boubenec<br>Jennifer Lawlor<br>Shihab Shamma<br>Bernhard Englitz |
| Advanced European Research Council | ERC 295603 | Yves Boubenec<br>Jennifer Lawlor<br>Shihab Shamma<br>Bernhard Englitz |
| European Commission's Marie Curie grant | 660328 | Shihab Shamma |
| Army Research Office | 63113-LS | Shihab Shamma |

The funders had no role in study design, data collection and interpretation, or the decision to submit the work for publication.

## Author contributions

YB, Conceptualization, Data curation, Formal analysis, Supervision, Validation, Investigation, Visualization, Methodology, Writing—original draft, Project administration, Writing—review and editing; JL, Data curation, Formal analysis, Validation, Investigation, Visualization, Methodology; UG, Data curation, Validation; SS, Funding acquisition, Validation, Writing—review and editing; BE, Conceptualization, Data curation, Formal analysis, Supervision, Funding acquisition, Validation, Investigation, Visualization, Methodology, Writing—original draft, Project administration, Writing—review and editing

## Author ORCIDs

Yves Boubenec, http://orcid.org/0000-0002-0106-6947
Jennifer Lawlor, http://orcid.org/0000-0002-6116-3001

## Ethics

Human subjects: All experiments were performed in accordance with the guidelines of the Helsinki Declaration. The Ethics Committees for Health Sciences at Université Paris Descartes approved the experimental procedures.

# Additional files

### Supplementary files

• Supplementary file 1. Example sounds embedding a change at 3s. The overall duration of the 4 stimuli is 5 s. Change size is 50%.

• Supplementary file 2. Same than *Supplementary file 1*, with change size 80%.

• Supplementary file 3. Same than *Supplementary file 1*, with change size 110%.

• Supplementary file 4. Same than *Supplementary file 1*, with change size 140%.

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
