## [Decision Letter]

[Editors’ note: a previous version of this paper was rejected after a second round of review, but the authors submitted for reconsideration. What follows is the first decision letter after peer review, requesting revisions.]

Thank you for submitting your article "Detecting changes in dynamic and complex acoustic scenes" for consideration by *eLife*. Your article has been reviewed by three peer reviewers, one of whom, Barbara G Shinn-Cunningham (Reviewer #1), is a member of our Board of Reviewing Editors, and the evaluation has been overseen by Timothy Behrens as the Senior Editor. The following individuals involved in review of your submission have agreed to reveal their identity: Jennifer K Bizley (Reviewer #2) and Simon P Kelly (Reviewer #3).

The reviewers have discussed the reviews with one another and the Reviewing Editor has drafted this decision to help you prepare a revised submission.

Summary:

This manuscript combines psychophysical, electrophysiological and modeling approaches to understanding auditory change detection. Listeners need to first estimate the probability distribution across a "tone cloud," then detect a change in this probability distribution. Behavioral data show that changes are easier to detect when they are later, larger, and when the spectral changes are in neighboring frequency channels. The authors complement these behavioral results with EEG data from a very similar task. Finally the paper includes two different models that account for the key characteristics of the behavioral data.

All three reviewers think the paper is interesting and publishable, but we had a number of concerns about the presentation. We appreciate that you had to make some difficult analysis decisions, both with respect to the psychophysics and the ERPs, and see that your choices are well considered and principled. The experimental results are compelling, showing nice behavioral and neuroelectric imaging on similar tasks. However, the modeling is not all that convincing, and the rationale for including both a statistical decision model and a more physiologically motivated model is not clear. We also have a fairly long list of minor edits / suggestions for you to consider to improve the clarity and readability of the manuscript.

Essential revisions:

1) The writing makes it hard to appreciate the study.

The manuscript is a big of a tough haul; it is quite heavy going and descriptive. The paper would be easier to read, and have bigger impact, if it were streamlined to better emphasize the research questions and hypotheses and to provide clear interpretations (e.g., to highlight what is learned from each experiment, to explain the importance of model parameter values). For instance, even understanding what the stimuli were and what the listeners were being asked to do was a real challenge. The description of the stimulus design is quite lengthy-yet very hard to follow. Perhaps it would help to start with a more intuitive, descriptive explanation of *what* you did, *why* you did it, and what the task was measuring before (in sentence two of your Stimulus Design section) jumping directly to descriptions of probabilities and marginal distributions and what the exact range of frequencies was. There is no forest emerging from these technical-tree details.

2) We also had concerns with the modeling.

2a) Claims about temporal dynamics of neural decisions that are based on averages of electrical activity over many trials are problematic. When you find a slow effect in an average, there is no way of knowing if activity is slowly accumulating within each trial, or if it is suddenly jumping from low to high at some point in time, but at different times, within each trial. In either case, you end up with an average that will look like a slow increase (see the recent paper by Jonathon Pillow on this topic). Many of the arguments about evidence accumulation rely on this aspect of the data. This weakness needs to be acknowledged and discussed.

2b) Assuming you can justify the accumulation modeling (see 2a), you need a clearer motivation for including *both* models and a better discussion of how the two relate to each other. Both offer some sort of insight into how this task might be solved, and can account for important characteristics of the dataset (including features that they were not explicitly modeled to fit). But it is not clear how the results of the cortical model relate to the accumulation dynamics. The cortical model seems to say that everything can happen in A1. But there is not any temporal integration function applied beyond A1, which seems discrepant with the CPP finding (unless the CPP comes from A1?). You address this briefly in the discussion, suggesting that the job of the CPP might be just to select and amplify certain accumulators, but these claims must be made more carefully with reference to the existing literature about the CPP signal. Because it is not clear how the two models are related, it is very hard to figure out what the take-away message is (e.g. how does the cortical model relate to the slow and fast approximation steps the statistical model?). What do we learn from having both? Do either (or both) generate testable predictions for further study? What experiments might be necessary to gather evidence for the statistical model's physiological instantiation? Either choose one model, which emphasizes what you want the reader to understand, or make clear how the models relate, and what each contributes to understanding the phenomena you observed.

3) There are some experimental points that need to be clarified.

The psychophysical and EEG sessions differed in potentially crucial ways, yet this is played down in the paper. The behavioral data from the EEG sessions are not reported at all. Comparing the psychophysics sessions to the EEG sessions, respectively: 1) responses are immediate versus delayed/withheld until stimulus offset, 2) there are 0 versus 50% catch trials, and 3) feedback is given versus not. These factors muddy the degree to which measured EEG signals reflect mechanisms and strategies for performing the task in the psychophysics sessions. We think it would help if you a) show the RT and accuracy (on both change and catch trials) data for the EEG session, and b) discuss these differences head-on, clarifying why it can be assumed that despite the methodological differences, the EEG effects reflect the same mechanisms at play during the psychophysics sessions.

4a) Analysis of the EEG data also raised some questions.

You conclude that neural activity in auditory cortex and parietal cortex carry different signals. However, since two completely different analyses were performed for auditory ERPs and the CPP, is this justifiable? The data in Figure 5 shows how the different analysis methods affect the resulting signals (the CPP are smoothed by the low-pass filtering). Are the differences between brain regions maintained if the same analysis methods are applied to both sets of electrodes? In order to determine that a decision-making signal is present in one set and not the other, shouldn't the same analysis be applied to both? Another example of this kind of issue is in subsection “EEG recordings and the site of decision-making”, where you conclude that the representation in the CPP electrodes is less temporally precise than the auditory cortical one. However, since the CPP signal has been low-pass filtered at 30 Hz, this is hardly surprising (!).

4b) Your "scene" may consist of a single, changing auditory object.

A "scene" typical has multiple "objects" that come and go; we question whether your stimuli are perceived as containing objects that appear and disappear. Superficially, this may seem like a semantic issue, but if one believes in "auditory objects" are the "units of attention" (as at least one of the reviewers does), this distinction has implications for the way in which the task is solved. Specifically, your task may depend on detecting changes in the spectral envelope of a single object, which could explain why you do not see change-related signals in auditory cortex, although other researchers do. For example, in the Chait lab, where change-related cortical signals are observed, acoustic scenes likely are perceived as consisting of model multiple objects that appear or disappear. If your listeners do hear multiple sources, this needs to be explained to the reader; if they hear one object changing, then this needs to be explained, and the discussion of how your study relates to "the real world" modified.

[Editors’ note: what now follows is the decision letter after re-review.]

Thank you for submitting your work entitled "Detecting changes in dynamic and complex acoustic environments" for consideration by *eLife*. Your article has been reviewed by two peer reviewers, and the evaluation has been overseen by a Reviewing Editor and a Senior Editor. The following individuals involved in review of your submission have agreed to reveal their identity: Jennifer K Bizley (Reviewer #2) and Simon P Kelly (Reviewer #3).

Our decision has been reached after consultation between the reviewers. Based on these discussions and the individual reviews below, we regret to inform you that your work will not be considered further for publication in *eLife*.

All of us are very interested in the topic and the questions you pose, and we all feel that the study has significant value, which is why we were receptive to the initial submission.

However, the new information provided in the revised submission brought to light problems with the data and interpretation that were not clear in the initial submission (see detailed reviews below). The corrected topographic plots in particular call into question the explanation of the activity as being attributable to an accumulation-based decision process. Unfortunately, the data included in the current manuscript do not provide sufficient support for your conclusions, and are inconsistent with published results on this topic (an issue that is not addressed in interpreting your results). The study would likely benefit from collection of additional data, as indicated below.

*Reviewer #2:*

The authors have addressed all of the concerns I originally outlined. I still have some reservations about the levels of performance of the subjects for all but the largest change sizes. Perhaps I've misunderstood the additional data included in Figure 5—figure supplement 3) on performance during the EEG task, but it seems that performance for catch trials and non-catch trials is indistinguishable suggesting that subjects are effectively randomly responding.

*Reviewer #3:*

I found that the authors were generally attentive to our comments, but an important part of the findings remains critically unconvincing. My own main issue was that it was not clear that the results of the EEG experiment really provide a view on the neural dynamics at play in the psychophysics task, because so many aspects of the task (immediate versus delayed response, presence of 50% catch trials, different set of change sizes and timings, and provision of feedback) were different.

Having prompted the authors to add the behavioral results for the task version run during EEG recording (which they did) and explain and discuss the impact of the task differences, I regret to say my concerns over this have grown rather than diminished. The authors state that "it is the same task" but it most certainly is not from the point of view of decision mechanisms, into which the study is supposed to provide new insights – the subjects listen out for the same kind of acoustic event, yes, but the various contingencies of any given task protocol – what they know or can surmise about timing, about the probability of the event happening at all, about the range of trial difficulties possible and certainly the requirement to report their detection immediately – can each have potentially huge impact on the strategies employed to perform the task and by extension the neural dynamics of decision formation.

The authors gave a single reason for the task differences: to avoid motor preparation signals in the EEG. This rationale does not hold up because the prior work identifying the "CPP" as a decision signal clearly demonstrated that response preparation can be easily dissociated from the CPP evidence accumulator simply from the fact that the CPP does not care about response hand. In fact, immediate decision reports were crucial because the ability to time-lock the signal to these actions and show that the accumulator peaks at response regardless of RT was one of the major identifying criteria of the decision signal.

Another major reason for my fallen enthusiasm is the fact that, after correcting an electrode-ordering mistake in this revision, the topography of the positive buildup appears to be focused at the very inferior edge of the cap. Neither the previously characterized decision signal "CPP" nor the P300 to which it has been equated, to my knowledge, has ever been found with such an inferior topography. The overall problem is that the authors are using the presence of this ERP positivity and the effects of change time on its buildup as evidence supporting their claim that this task is accomplished through sensory evidence accumulation. But for this to work, it needs to be definitively and independently established that signal is in fact an evidence accumulator – otherwise, the argument is circular, i.e., the neural evidence for the improved change detection at later change times being a result of steeper evidence accumulation comes from the finding that an accumulator signal rises more steeply for later change times, but the sole basis upon which that signal is identified as an evidence accumulator is the fact that it rises more steeply for later change times! Clearly what is needed is an independent way to establish that the signal is an evidence accumulator. Shared properties with an established accumulator signal (CPP) gave at least partial support, but with the topography now so distinct I think the authors have lost this one piece of support. Source localizing to parietal cortex does not help much because parietal cortex in general has not been associated with decision formation any more than any other cognitive function, let alone with evidence accumulation in particular, and a parietal neural source certainly played no part in the CPP's identification as a decision signal.

I feel my hands are tied on highlighting this issue as a preclusive one when I consider the simple fact that if an ERP study with as few as 6 subjects, equating an occipital positivity with a previously characterized centro-parietal one, were to be submitted to a low-ranking but specialized journal like Psychophysiology, it would likely be dismissed out of hand.

The obvious solution to all of the above would be to collect EEG data from a more reasonable number of subjects (>15) on a task with identical parameters to that run in the psychophysical sessions. This would readily allow independent tests of the assertion that the positivity is an evidence accumulator: 1) The peaking at RT regardless of whether RT is fast or slow I mentioned before, and 2) The other major identifying property is the dependence of buildup rate on evidence strength which could be tested across the four different change sizes 50-140%. All the authors would have to do to deal with the broader range of change times is to sort trials into change-time bins. In addition to these independent tests, more reliable analyses could be conducted to address many other unanswered questions in the paper about what the positive buildup signal is actually doing in the task, e.g., what temporal profile does the signal have during the baseline pre-change part of the stimulus – does it steadily build up reflecting the accumulation of information used to construct the baseline statistical distribution, or does the signal step up discretely as suggested by the current waveforms, and if so, why? And what is the signal actually reflecting with respect to the models, e.g. the sum of the two accumulators working on different timescales? Are the apparent buildup rate effects seen already in the pre-change baseline for later changes statistically reliable, and if so what could this mean?

Further comments

The visual topography of the positivity: The authors measure the positive signal from electrodes Pz to Oz on the posterior midline. Pz is the most inferior the CPP has been measured thus far (Twomey et al. 2015, cited by the authors), and this is included in the authors' cluster, but where is the signal actually biggest? From the topography it looks like it could be Oz, and this strangely suggests a visual cortical origin. It now becomes important to know what the subjects actually looked at during the task, which is not detailed anywhere – did they fixate on a cross and nothing else? Were there stimuli on the screen that were presented during the trial or after, and could be processed or anticipated by the subject? One alternative possibility to a decision buildup account is that this positivity reflects suppression of visual cortex upon detection of the change, which would happen earlier on late-change trials where the change is detected quickest.

If the occipital positivity is in fact the same signal as the CPP, then the reason for its dramatic inferior shift should be figured out. A likely reason is that 6 subjects simply don't provide a reliable enough measurement, and one or two outlying subjects are dominating. But it could also be a signal issue – maybe the polynomial detrending has removed the actual decision signal from centro-parietal sites? "nt_detrend" probably has been used exclusively for datasets with relatively fast responses, and not for tasks where the potentials are very slow-moving like the current one – what happens to the signals when this step is removed, or replaced by a first-order (linear) detrend? It is possible in principle that strong auditory-evoked negative activity more frontally causes the positive focus to shift back on the scalp, but this seems an unlikely explanation given that the CPP/P300 have been measured with the same centro-parietal topographies for many auditory tasks including ones containing continuous audio streams (O'Connell et al. 2012).

Abstract: "in parietal cortex" should definitely be replaced with "over parieto-occipital scalp". This is EEG.

[Editors’ note: what now follows is the decision letter after the authors resubmitted for consideration.]

Thank you for resubmitting your work entitled "Detecting changes in dynamic and complex acoustic environments" for further consideration at *eLife*. Your revised article has been favorably evaluated by Timothy Behrens (Senior and Reviewing editor), and two reviewers.

The manuscript has been improved but there are some remaining issues that need to be addressed before acceptance, as outlined below:

*Reviewer #1:*

In this paper, Boubenec and colleagues demonstrate compelling psychophysical and electrophysiological correlates of statistical estimation mechanisms underlying naturalistic acoustic change detection, and provide both an abstract and physiologically-based computational model to explain their data. This is a revision of a previous submission, based on new EEG data recorded during the exact same task as was physiologically characterized and modeled, and the results are now more definitive and highly interesting, including stronger evidence for accumulation (steeper buildup for stronger changes) and additional effects on peak amplitude of the neural index of accumulation. The concepts and hypotheses are now also laid out very clearly, with nice sections explaining predictions based on statistical estimation strategies. There are just a couple of substantive points that are important to clear up at this stage.

There are a couple of important checks to perform on the CPP amplitudes at response to verify the effects on this measure. First, the topography of the effect of change size on amplitude should be shown, around the time of response, e.g. the largest change size minus smallest – does it have a similar CPP topography? This would go some way to verifying that the amplitude effects are not caused by an overlapping, separate process (e.g. slow fronto-central negativity). Also, since the baseline for the response-locked waveforms was 1000-1200ms prior to response, conditions with longer RTs are baselined relative to intervals further out into the stimulus; this is a potential confound because it could be that for lower change sizes, the hit trials are those in which the CPP waveform gets a head-start by accumulating noise or a false change, and so it has less of a distance to go to threshold when the actual change happens. Or given how long some RTs are for weaker changes, the "head-start" could be from genuine physical evidence! This would result in an amplitude decrease at response when using the current baselining regime, even in the absence of any change in bound. Therefore a pre-stimulus or a pre-change baseline should instead be used for the response-locked waveforms. It is important to see whether this changes the results. I would be fairly confident that the main effects hold up – the slope effects certainly will because baselining doesn't matter in that case. Nevertheless, these controls are important to rule out critical potential confounds.

Possible explanations for the decrease in amplitude of CPP across change times tend to be provided briefly in a scattered fashion. It is important to reiterate the lack of effect of RT here – if bound is collapsing over time, surely the CPP amplitude would decrease with RT as well as with change time? Further, as the authors point out themselves, if bound were collapsing over time, false alarm rate should increase – it seems that this point should be weaved into the discussion of this effect more than it is. In general, since several possible explanations are given for this effect, it might be better not to scatter them in various parts of the Results and Discussion, but rather go through them all in one coherent section in the Discussion?

Around paragraph three of subsection “EEG recordings and the site of decision-making”the discussion is a bit muddled – it appears to both argue for and against a decreasing bound in the same breath. This could use clarifying.

Finally on this point, have the authors considered other, simpler mechanisms for preventing the subject from clicking when the stimulus initially onsets? The work of Tobias Teichert (see e.g. review in 2014) and others have demonstrated that subjects may be in control of the timing of onset of decision processes (specifically evidence accumulation) – given there's a "dead zone" apparent in Figure 1 showing that subjects are not willing to respond until a certain time has elapsed, couldn't they be simply deferring the kick-off of their decision mechanisms? This is easily parameterized in a fixed delay, and would avoid the need to introduce dramatic time-dependence in the fixed mechanisms (e.g. lengthening of the slow estimator's time constant) to explain that aspect. The authors should at least discuss this possibility.

Figure 5 think it might make more sense to have each signal (medial, CPP) in the rows, and the response-locked waveforms to the right of the stimulus locked ones, as is standard across neurophysiological decision research. The labels should also be consistent in nature – perhaps they should be "fronto-central" and "parietal?" – "medial" is not a good descriptor since both signals are medial.

Missing from Figure 5 main figure are the change-onset locked waveforms. Supplement 3 shows the auditory electrodes, but really both signals should be shown time locked to change onset for a complete view over the dynamics. The x-axes of the current figures can easily be cut to zoom in on the main dynamics (e.g. the current 500-ms baseline shown before stimulus onset need only be 100 ms), so there should be room to squeeze them in between. This is important given that in places there are references to what happens to the CPP in response to the change, not just prior to response.

I believe *eLife* format requires methods to go at the end.

*Reviewer #2:*

The inclusion of the additional data (18 subjects in a simultaneous behavioural-EEG paradigm) significantly strengthens this manuscript providing clarity on the issues raised in previous reviews. The manuscript makes a significant contribution to the auditory field and I have only a few minor comments.

---

## [Author Response]

[Editors’ note: the first author response to the requests for revisions follows.]

*Essential revisions:*

1) The writing makes it hard to appreciate the study.

*The manuscript is a big of a tough haul; it is quite heavy going and descriptive. The paper would be easier to read, and have bigger impact, if it were streamlined to better emphasize the research questions and hypotheses and to provide clear interpretations (e.g., to highlight what is learned from each experiment, to explain the importance of model parameter values). For instance, even understanding what the stimuli were and what the listeners were being asked to do was a real challenge. The description of the stimulus design is quite lengthy-yet very hard to follow. Perhaps it would help to start with a more intuitive, descriptive explanation of* what *you did,* why *you did it, and what the task was measuring before (in sentence two of your Stimulus Design section) jumping directly to descriptions of probabilities and marginal distributions and what the exact range of frequencies was. There is no forest emerging from these technical-tree details.*

The description of the stimulus is inherently complex, and so we agree with the concern that the description in Methods does not offer an 'easy way in'. We therefore added more entry text to give a simpler, intuitive description ahead of the stimulus & procedure description in the Methods (paragraph Stimulus Design & Trial Procedure). We also added a set of sound samples as supplementary material, using the possibility of embedding multimedia files within the course of the paper. Also, we have revised the Abstract, Introduction and Discussion to streamline the presentation for improved readability.

*2) We also had concerns with the modeling.*

2a) Claims about temporal dynamics of neural decisions that are based on averages of electrical activity over many trials are problematic. When you find a slow effect in an average, there is no way of knowing if activity is slowly accumulating within each trial, or if it is suddenly jumping from low to high at some point in time, but at different times, within each trial. In either case, you end up with an average that will look like a slow increase (see the recent paper by Jonathon Pillow on this topic). Many of the arguments about evidence accumulation rely on this aspect of the data. This weakness needs to be acknowledged and discussed.

Indeed, Jonathan Pillow and colleagues recently showed that the spiking activity of individual neurons in the parietal areas could follow discrete and instantaneous change in their underlying firing distribution. However, as far as we know, they do not suggest that simultaneous discrete steps would occur at the level of the population. Rather, the idea is that each neuron will independently undergo a discrete change in its spiking activity at different times from one trial to the next. Thus, at the single trial level, one would expect a ramping activity if one is to average responses from a large neuronal population, to produce a ramping response much like that of the cortical model or the EEG recordings. To clarify this point further to the readers, we added the following statements in the EEG recordings and the site of decision-making paragraph of the Discussion section:

“It has recently been suggested that individual neurons change their firing rate instantaneously at the single trial level (Latimer et al., 2015). We presently observed gradual, rather than step-wise changes in our across-trial averages. However, we predict that even single trial EEG signals would be gradual as these step-changes occur randomly, and hence are unlikely to be synchronized at the population-level. Due to the large ensemble of neural responses contributing to a single scalp location’s potential, this instead results in the commonly seen ramping activity on the EEG level, as observed in our data.”

*2b) Assuming you can justify the accumulation modeling (see 2a), you need a clearer motivation for including* both *models and a better discussion of how the two relate to each other. Both offer some sort of insight into how this task might be solved, and can account for important characteristics of the dataset (including features that they were not explicitly modeled to fit). But it is not clear how the results of the cortical model relate to the accumulation dynamics. The cortical model seems to say that everything can happen in A1. But there is not any temporal integration function applied beyond A1, which seems discrepant with the CPP finding (unless the CPP comes from A1?). You address this briefly in the discussion, suggesting that the job of the CPP might be just to select and amplify certain accumulators, but these claims must be made more carefully with reference to the existing literature about the CPP signal. Because it is not clear how the two models are related, it is very hard to figure out what the take-away message is (e.g. how does the cortical model relate to the slow and fast approximation steps the statistical model?). What do we learn from having both? Do either (or both) generate testable predictions for further study? What experiments might be necessary to gather evidence for the statistical model's physiological instantiation? Either choose one model, which emphasizes what you want the reader to understand, or make clear how the models relate, and what each contributes to understanding the phenomena you observed.*

The dual-timescale model, which accounts well for the data, is clearly only remotely related to the physiology, and instead mostly applies at the level of statistical estimation as a guiding principle. Since Drift-Diffusion Models showed a great success in describing many experimental data, we think it is important to demonstrate that our psychophysical data can be explained by such a mechanism. On the other hand, the cortical model simulations make a good case for a consistency of a lack of grand response in A1 and the local processing in the model.

We have now added text to motivate the choice of the models and their relative strengths. We also described in more detail the “mechanistic” differences of how the two models accomplish their performance, and on which level we would see their implementation. Briefly, detection of statistical changes can be accomplished by a classical statistical estimation model, but can also be accounted for by a different, more physiological model with multiple time-scales. We made this idea easier to understand for the reader with the following text (Modelling statistical decision-making on two levels paragraph of the Discussion section):

“To create behavioral performance from its representation, we merely added a filter selection and a decision criterion. The spectrotemporal filters implemented in the cortical model exhibit alternating excitatory (positive) and inhibitory (negative) fields (Figure 8) that compare the spectral stimulus properties over a given time window set by a filter's temporal rate. As such, it effectively integrates the recent input with opposite signs to detect a change, which can be compared to the difference between the fast and slow estimators in the statistical estimation model.”

3) There are some experimental points that need to be clarified.

*The psychophysical and EEG sessions differed in potentially crucial ways, yet this is played down in the paper. The behavioral data from the EEG sessions are not reported at all. Comparing the psychophysics sessions to the EEG sessions, respectively: 1) responses are immediate versus delayed/withheld until stimulus offset, 2) there are 0 versus 50% catch trials, and 3) feedback is given versus not. These factors muddy the degree to which measured EEG signals reflect mechanisms and strategies for performing the task in the psychophysics sessions. We think it would help if you a) show the RT and accuracy (on both change and catch trials) data for the EEG session, and b) discuss these differences head-on, clarifying why it can be assumed that despite the methodological differences, the EEG effects reflect the same mechanisms at play during the psychophysics sessions.*

The differences between the psychophysical and EEG sessions are now discussed explicitly right after their introduction in the Materials and methods.

“The delayed response and catch trials were introduced in the EEG study to improve the reliability of observing the neural response. Delaying the response allows us to observe the neural integration, without a (preparatory) motor response interfering. Since listeners had more of an opportunity to respond in each trial, catch trials had to be introduced to assess detection performance (see Figure 5—figure supplement 1). While the overall set of trials differed somewhat between the psychophysics and the EEG experiments, the task in both cases is the same. Given the (on average) longer integration duration before a decision is made in the EEG, we expected reaction times and performance to differ less. However, they were both significantly dependent on change time (Figure 5—figure supplement 1).”

Further, a figure supplement has been added to Figure 5—figure supplement 1), which details both reaction times and performance for the EEG condition. Both reaction times and performance show a dependence on change time, although the absolute reaction times are considerably longer than in the psychoacoustic version, since subjects had less pressure to react quickly. Subjects appeared to be conservative for short trials, i.e. did not detect a change in statistics, and hence were at 'below-chance' in change detection, but above-chance 'catch detection'.

4a) Analysis of the EEG data also raised some questions.

*You conclude that neural activity in auditory cortex and parietal cortex carry different signals. However, since two completely different analyses were performed for auditory ERPs and the CPP, is this justifiable? The data in Figure 5 shows how the different analysis methods affect the resulting signals (the CPP are smoothed by the low-pass filtering). Are the differences between brain regions maintained if the same analysis methods are applied to both sets of electrodes? In order to determine that a decision-making signal is present in one set and not the other, shouldn't the same analysis be applied to both? Another example of this kind of issue is in subsection “EEG recordings and the site of decision-making”, where you conclude that the representation in the CPP electrodes is less temporally precise than the auditory cortical one. However, since the CPP signal has been low-pass filtered at 30 Hz, this is hardly surprising (!).*

This is an excellent point that we checked beforehand but did not include in the manuscript at that time. We now inserted a figure supplement to Figure 5 showing the low-pass filtering analysis applied to the set of auditory electrodes. The buildup signal is specific to the parieto- occipital electrodes as shown by the absence of any post-change significant activity over the low-passed auditory-related electrodes. We also refer to this supplement figure in the manuscript:

“This signal was absent from the set of auditory electrodes when applying the same analysis (Figure 5—figure supplement 2), indicating that this post-change activity was specific to the group of parieto-occipital electrodes.”

We also realized that a minor part of the electrode labels were actually shifted by 1 index. This does not affect qualitatively the previous results. Auditory electrodes were untouched by this correction whereas the electrodes displaying the slow post-change potential were more occipital than what we described in the previous version of the manuscript (see the updated topographic plot in Figure 5). This has been corrected in the entire revised manuscript.

In order to verify that this occipital shift of the potential is not at odds with the CPP signal stemming from a parietal source, we performed source localization on the onset and change components (using the clustering approach in the dipfit toolbox in EEGLAB). For the onset component of the response, associated with the fronto-central positivity (Figure 5—figure supplement 3 A and B) the dipoles are localized in superior temporal cortex, BA 41 (Figure 5—figure supplement 3C). For the change component of the response, the dipoles are more distributed but cluster around the parietal cortex, in the range beyond TPJ (temporal parietal junction). We are aware that source localization is not trivial, however, the present locations were estimated without much tweaking of parameters, and are cleanly associated with the temporal response shapes we discussed in the central and the parieto-occipital electrode locations.

4b) Your "scene" may consist of a single, changing auditory object.

*A "scene" typical has multiple "objects" that come and go; we question whether your stimuli are perceived as containing objects that appear and disappear. Superficially, this may seem like a semantic issue, but if one believes in "auditory objects" are the "units of attention" (as at least one of the reviewers does), this distinction has implications for the way in which the task is solved. Specifically, your task may depend on detecting changes in the spectral envelope of a single object, which could explain why you do not see change-related signals in auditory cortex, although other researchers do. For example, in the Chait lab, where change-related cortical signals are observed, acoustic scenes likely are perceived as consisting of model multiple objects that appear or disappear. If your listeners do hear multiple sources, this needs to be explained to the reader; if they hear one object changing, then this needs to be explained, and the discussion of how your study relates to "the real world" modified.*

We agree with the reviewers that the choice of terminology was not optimal. Auditory scenes are classically considered to have a small number of identifiable acoustic objects, whereas acoustic textures are better categorized under the term 'acoustic environment', in analogy with the typical, naturally occurring examples of auditory textures, i.e. rain, wind, water, etc, typical for natural environments. From this perspective, we agree – and never intended to claim anything different – that the auditory texture is typically perceived as a whole, while being composed of a large number of acoustic elements (e.g. drops, bubbles, short tones in our case). While listeners may differ in their perception/strategy, each could choose to listen to a certain subset of the texture, e.g. certain constituents or – as suggested in point 10 by the reviewers – a certain frequency range, e.g. the loudest one. We show in the new Figure 2—figure supplement 3, that the latter strategy is unlikely, given the performance data, see point 10 for details. If a listener had, nonetheless, adopted such a strategy, we hypothesize that auditory objects would be transiently formed on an individual element basis, but it would still hardly qualify as an acoustic scene. In consequence, we have gone through the manuscript and replaced the term 'scene' with 'environment', and carefully scouted for similar formulations.

Regarding the work of Prof. Chait, we have devoted a new paragraph in the Discussion to a comparison with her work.

[Editors’ note: the author responses to the second round of review (rejection) follows.]

*Reviewer #2:*

*The authors have addressed all of the concerns I originally outlined. I still have some reservations about the levels of performance of the subjects for all but the largest change sizes. Perhaps I've misunderstood the additional data included in the supplemental material Figure 5—figure supplement 3) on performance during the EEG task, but it seems that performance for catch trials and non-catch trials is indistinguishable suggesting that subjects are effectively randomly responding.*

The task was designed with a high level of difficulty, to ensure that subjects had to listen closely to detect a change. We think that the performance data is indicative of the subjects actively trying to solve the task, rather than guessing, for the following reasons:

– The performance depends strongly and significantly on the change time. Hence, both above 50% performance in the 3s condition and below 50% performance for the 0.75s condition are indicative of actively performing subjects, otherwise chance performance would be expected for both. As in other detection paradigms, some conditions are easy, leading to above chance performance, whereas others are designed to be very hard to detect. Because conditions were not presented in blocks, subjects could not be biased towards chance performance by balancing their perceptual reports to equate the change and no change responses within condition. This led to below chance performance for the hardest no-catch condition (50%) where subjects were missing a substantial proportion of change. If we were to continue the change time axis in both directions, we would expect to find a sigmoidal dependence, possibly not actually reaching 0 and 100% due to fatigue and unsystematic response errors. However, the average performance of these subjects across all non-catch conditions could well be 50%, however, the performance is systematically determined by the independent variable (change time here). We agree, that a higher level of performance would be more reassuring about the subject's strategy in all conditions, however, this would come at the cost of making the task easy for every parameter value, and thus losing the dependence of performance on variables of the task.

– Certainty of decision appeared to vary as a function of change time as well, since the reaction times for both catch and signal trials reduced significantly with change time. This indicates also that subjects were not guessing after stimulus end, but made an active decision based on the previous information.

– The correct classification of catch trials ('no response') overall is significantly greater than chance (p=0.0000007, t-test of all conditions compared to 50%). This is the performance indicated by the percentages (y-axis) in B. The slight decrease in performance with change time in this bigger dataset surprised us, but could indicate an increased sensitivity to small fluctuations over time, which then erroneously classify a subset of the catch trials as signals.

Author response image 1.Change detection reaction times and performance during the delayed response EEG experiment as a function of exposure to the first texture Reaction time decreased significantly as a function of change time and trial type both for catch (brown) and change trials (blue, 1 way ANOVA, p-values indicated in the figure).Reaction times were normalized within each subject before averaging to account for individual overall differences. (A) The accuracy (correct response for either trial type) of catch trials stayed unchanged (brown, 1-way ANOVA), while the performance for the change trials improved significantly with change time (blue, 1-way ANOVA).**DOI:**
http://dx.doi.org/10.7554/eLife.24910.020

Please note that this plot is not provided in the manuscript, since in response to reviewer 3, the delayed response dataset has been replaced with the “immediate” response dataset. The performance in that task (during EEG performance) is depicted in Figure 5—figure supplement 1.

*Reviewer #3:*

*I found that the authors were generally attentive to our comments, but an important part of the findings remains critically unconvincing. My own main issue was that it was not clear that the results of the EEG experiment really provide a view on the neural dynamics at play in the psychophysics task, because so many aspects of the task (immediate versus delayed response, presence of 50% catch trials, different set of change sizes and timings, and provision of feedback) were different.*

*Having prompted the authors to add the behavioral results for the task version run during EEG recording (which they did) and explain and discuss the impact of the task differences, I regret to say my concerns over this have grown rather than diminished. The authors state that "it is the same task" but it most certainly is not from the point of view of decision mechanisms, into which the study is supposed to provide new insights – the subjects listen out for the same kind of acoustic event, yes, but the various contingencies of any given task protocol – what they know or can surmise about timing, about the probability of the event happening at all, about the range of trial difficulties possible and certainly the requirement to report their detection immediately – can each have potentially huge impact on the strategies employed to perform the task and by extension the neural dynamics of decision formation.*

While we think both task designs have their merits, we agree that an exact match between the paradigms removes potential doubts. We would like to thank the reviewer for this criticism, prompting us to add the data from the immediate response experiment. Here, subjects to respond immediately and the stimulus conditions are matched (change times and change sizes) with the psychophysical experiment (N=18, depicted in the new Figure 5, and its figure supplements). The data from the delayed paradigm have become unnecessary, and its presentation would complicate the presentation to a degree that would not be in the interest of the readers. We have therefore replaced the corresponding figures and text with the new, matched task. Below we refer to the delayed task a few times, to address some of the reviewer questions. Further details are provided below.

*The authors gave a single reason for the task differences: to avoid motor preparation signals in the EEG. This rationale does not hold up because the prior work identifying the "CPP" as a decision signal clearly demonstrated that response preparation can be easily dissociated from the CPP evidence accumulator simply from the fact that the CPP does not care about response hand. In fact, immediate decision reports were crucial because the ability to time-lock the signal to these actions and show that the accumulator peaks at response regardless of RT was one of the major identifying criteria of the decision signal.*

We agree that the motor response may not have a disruptive influence on the CPP potential or invalidate its significance as a decision signal. However, the fact that the motor response is synchronous with parts of the CPP signal may lead to shifts in topography and changes in timing (see e.g. Salisbury et al., 2001). We think that it remains an interesting and important question to resolve where the difference in location between the post-change potentials arises from, however, this would require this manuscript to grow beyond the limits of the *eLife* format.

We also agree that the possibility to time-lock the EEG response to the behavioral response time is crucial for the analysis of decision signals. For the purpose of the present manuscript, we can confirm that in the immediate response task, the location of the response-locked potential is consistent with previous CPP reports. Within the added dataset (immediate response, matched conditions) we provide these time-locked analyses (new Figure 5 and figure supplements) and demonstrate that the present signal recorded in the CPP location exhibits similar properties to the signals recorded by the reviewer and others in evidence accumulation tasks (see below, point 4, for more details). In addition we show a decrease in the potential as a function of change time – to our knowledge a novel finding – which could be indicative of a reducing threshold as a function of time-into-trial.

*Another major reason for my fallen enthusiasm is the fact that, after correcting an electrode-ordering mistake in this revision, the topography of the positive buildup appears to be focused at the very inferior edge of the cap. Neither the previously characterized decision signal "CPP" nor the P300 to which it has been equated, to my knowledge, has ever been found with such an inferior topography. The overall problem is that the authors are using the presence of this ERP positivity and the effects of change time on its buildup as evidence supporting their claim that this task is accomplished through sensory evidence accumulation. But for this to work, it needs to be definitively and independently established that signal is in fact an evidence accumulator – otherwise, the argument is circular, i.e., the neural evidence for the improved change detection at later change times being a result of steeper evidence accumulation comes from the finding that an accumulator signal rises more steeply for later change times, but the sole basis upon which that signal is identified as an evidence accumulator is the fact that it rises more steeply for later change times! Clearly what is needed is an independent way to establish that the signal is an evidence accumulator. Shared properties with an established accumulator signal (CPP) gave at least partial support, but with the topography now so distinct I think the authors have lost this one piece of support. Source localizing to parietal cortex does not help much because parietal cortex in general has not been associated with decision formation any more than any other cognitive function, let alone with evidence accumulation in particular, and a parietal neural source certainly played no part in the CPP's identification as a decision signal.*

We can now address this point more encompassingly in light of the added dataset. We will first reiterate here the criteria for considering our observations as a decision signal, its relation to the CPP as well as to our present main question, the detection of changes in the statistics of auditory textures.

First, a useful set of criteria (or at least indicators) for a decision signal as provided by O'Connell et al., 2012, Kelly & O'Connell, 2013 and others, are

1) Encoding the integral of sensory evidence (i.e. linearly for constant evidence as a function of time or quadratically, if the amount of sensory evidence increases linearly itself, as in O'Connell et al., 2012).

a) As a corollary, the slopes of the potential should increase with the level of evidence.

2) Existence of a threshold, which determines when a sufficient amount of evidence has been reached that validates a response.

a) As a corollary, EEG responses for early and late reaction times should have similar height.

b) As suggested by a few models, the threshold could depend on time, indicating either an increase in certainty or relating to the task design (e.g. an expectation of a trial end)

c) The size of the EEG response at decision time (or just before) should not depend on the rate of evidence.

d) False alarms should have low/lower EEG responses than signal conditions, indicating that stochastically a decision was made before reaching the actual threshold (or a fluctuating threshold that is lower than the typical threshold)

3) Other properties would be generalizing over modalities, independence of response type, etc., which were not tested here.

4) In relation to previous reports on EEG potentials related to evidence integration, one would expect a centro-parietal location with a positive polarity, known as CPP, as first reported by O'Connell et al., 2012.

Based on the new dataset we can confirm essentially all of these points, with details provided below. Starting with point 4, the CPP was defined as the average potential from electrodes 14, 27 and 28 (EasyCap, 61 Channel, Equidistant layout, where for example electrode 14 has coordinates Theta = 45, Phi = 90) based on the peak region of the potential (see Figure 5). The potentials scalp location is hence consistent with previous reports (e.g. Twoney et al. 2015), although still slightly more occipital than some reports (O'Connell et al. 2012). These residual differences could be related to the complexity of the auditory stimulus, where a more complex stimulus (as presently used) may lead to a slightly more posterior signal from higher auditory cortex, which mixes with the generator of the CPP, or to some modality-specific changes in the localization of the generator (Dreo et al., 2016).

Considering point 1a above, we find that the slopes depend significantly on the change size, which in the present experiment can be equated with the instantaneous rate of evidence. Hence, the decision variable should increase linearly as a function of time, with increasing slopes for higher change sizes. We indeed find such a highly significant dependence (Figure 5).

Considering point 2a, we find the potentials for early and late reaction times to not be significantly different (see Figure 6).

Considering point 2b, we find the threshold to decrease as a function of change time, with the most significant decrease towards the end of the trial (Figure 5). This would be consistent with subjects expecting the end of the maximal length trial (which is not completely avoidable, even in the present case of an exponential distribution of change times), and adjusting their response threshold as they approach the expected end. However, we did not find an increase in the false alarm rate towards the end of the trial, which should have accompanied such a behavior. On the other hand, converging thresholds have also been suggested in other decision theoretic accounts (Bogacz et al, 2009). The reduction in threshold could, hence, reflect the improved estimate of the statistics of the first texture, thus requiring less evidence to make a decision at the same level of certainty.

Considering point 2c, we in fact find an increase of the CPP height with the change size, similar to what was previously reported (O'Connell et al., 2012). While this may appear to be inconsistent with a fixed threshold, we would like to suggest a possible interpretation: If the CPP potential is read off, and informed by the decision maker, then in the intervening time the CPP could continue to rise and thus achieve differential final levels for different rates of inst. evidence. Consistent with this idea, we find the CPP to not depend significantly on the change size in the window of 200-100 ms before the response time (2-way ANOVA, as in Figure 5 lower panels, only with a shifted time window).

Considering point 2d, we find that the CPP of false alarms is significantly smaller than the CPP for hits (p=1e-9, 1-way ANOVA as a function of change size), mirroring a result by O'Connell et al., 2012.

Together these results seem to indicate that the significance of the CPP as indicator for an evidence integrating, decision related process extends to the domain of complex broadband acoustic stimuli, such as acoustic textures.

Next, we would like to discuss the implications in the context of auditory texture processing, as well as the relation to our models. Different views exist on whether an auditory texture is mostly recognized based on its statistics (as has recently been proposed by McDermott and colleagues) or whether it is individual, characteristic elements that are indicative of the texture type. Within the limits of our paradigm, the identification of the CPP signal and its dependence on change size supports the hypothesis that evidence is (statistically) integrated, rather than instantaneously recognized.

The properties of the CPP, especially its relation to change size, are reflected in both models we proposed. The slope of the decision variables is in both cases increasing with the instantaneous evidence (this follows directly from the defining equations), and the height of the potential will depend on the combination of the instantaneous rate of evidence, i.e. on the change size. In both models a transient overshoot of the variable above the threshold would be observed, if the trials are terminated by the moment of decision execution, rather than internal evidence read-out, hence, the potential size would also depend on the change size.

Finally, the converging threshold used in the cortical model is reminiscent of the reduction in CPP size as a function of change time. An alternative interpretation of the reduction in threshold could, however, be the subjects roughly estimate the trial end, which could lead to a lowering of their threshold criterion. The constant FA rate (Figure 2), however, argues against this interpretation. We therefore favor the interpretation that the reduction in threshold is a consequence of an improved estimate of the statistics of the first texture, thus requiring less evidence to detect a deviation in statistics at the same level of certainty.

Reference:

Dreo, J., Attia, D., Pirtošek, Z. and Repovš, G. (2016), The P3 cognitive ERP has at least some sensory modality-specific generators: Evidence from high-resolution EEG. Psychophysiol. doi:10.1111/psyp.12800

*I feel my hands are tied on highlighting this issue as a preclusive one when I consider the simple fact that if an ERP study with as few as 6 subjects, equating an occipital positivity with a previously characterized centro-parietal one, were to be submitted to a low-ranking but specialized journal like Psychophysiology, it would likely be dismissed out of hand.*

We hope to have addressed these issues fully now.

The obvious solution to all of the above would be to collect EEG data from a more reasonable number of subjects (>15) on a task with identical parameters to that run in the psychophysical sessions. This would readily allow independent tests of the assertion that the positivity is an evidence accumulator: 1) The peaking at RT regardless of whether RT is fast or slow I mentioned before, and 2) The other major identifying property is the dependence of buildup rate on evidence strength which could be tested across the four different change sizes 50-140%. All the authors would have to do to deal with the broader range of change times is to sort trials into change-time bins.

As detailed above, we have confirmed both properties mentioned (1 & 2) on the new matched dataset (n=18), with the change-times divided into 4 bins.

*In addition to these independent tests, more reliable analyses could be conducted to address many other unanswered questions in the paper about what the positive buildup signal is actually doing in the task, e.g., what temporal profile does the signal have during the baseline pre-change part of the stimulus – does it steadily build up reflecting the accumulation of information used to construct the baseline statistical distribution, or does the signal step up discretely as suggested by the current waveforms, and if so, why? And what is the signal actually reflecting with respect to the models, e.g. the sum of the two accumulators working on different timescales? Are the apparent buildup rate effects seen already in the pre-change baseline for later changes statistically reliable, and if so what could this mean?*

We agree that we have so far not placed a significant emphasis on the build-up signal in the pre-change period (depicted in Figure 5 (old) before). The presence of this building-potential after stimulus onset is indeed interesting, although again its interpretation is not trivial: this potential is present for both the delayed response and the immediate response paradigms, however, only positive and dominant at the occipital electrodes (in both paradigms!). Looking at the new EEG data at the CPP electrodes (14,27,28) there is barely any deflection visible (Figure 5).

Hence, while it would be tempting to speculate about the meaning of the potential w.r.t. evidence integration, this would be better informed with a matched fMRI study, that can pinpoint the location of an integration/sensory memory (as e.g. in Linke et al., 2011). Along the lines of the cited study, it could also be interpreted as a persistent suppression in activity in sensory cortex, which is reflected in a dipole increase at another location, here the occipital electrodes.

Author response image 2.Recreated Figure 5 for the delayed paradigm with a larger number of subjects (n=13), demonstrating that the topography of the potential is unchanged, as are the dependence of slope on change time (which we, however, now interpret as a combination of change time and response time).**DOI:**
http://dx.doi.org/10.7554/eLife.24910.021

*Further comments*

*The visual topography of the positivity: The authors measure the positive signal from electrodes Pz to Oz on the posterior midline. Pz is the most inferior the CPP has been measured thus far (Twomey et al. 2015, cited by the authors), and this is included in the authors' cluster, but where is the signal actually biggest? From the topography it looks like it could be Oz, and this strangely suggests a visual cortical origin. It now becomes important to know what the subjects actually looked at during the task, which is not detailed anywhere – did they fixate on a cross and nothing else? Were there stimuli on the screen that were presented during the trial or after, and could be processed or anticipated by the subject? One alternative possibility to a decision buildup account is that this positivity reflects suppression of visual cortex upon detection of the change, which would happen earlier on late-change trials where the change is detected quickest.*

*If the occipital positivity is in fact the same signal as the CPP, then the reason for its dramatic inferior shift should be figured out. A likely reason is that 6 subjects simply don't provide a reliable enough measurement, and one or two outlying subjects are dominating.*

The screen remained static during the entire duration of sound presentation, i.e. there was no visual cue presented close to the time when the change occurred (this information has been added in more detail to the Methods section of the manuscript, thanks for pointing this out). With visual stimulation excluded, there are two main alternatives left: either (1) mixing with a motor signal, or (2) generation of a different signal.

1) Button presses have been shown to affect the P300 topography (e.g. Salisbury et al., 2001), however, a comparison with their results is difficult, since their P300 location was again different to start with. Their results, however, indicate that the location of a right-hand button press component is on the left side, which aligns with a slight leftward shift that we observe in our immediate response CPP potential (new Figure 5, aligned to response).

2) Regarding your hypothesis that the positivity reflects suppression of visual cortex: Given the consistent location of the change potential in the CPP location for the immediate responses, we think that – together with its properties – there is little doubt about the nature of the signal, at least in the immediate response case (see additional arguments above following your point 4). We think that in the delayed case, interpreting the shift to the more posterior location as a suppression of visual cortex is not likely, given that we have previously demonstrated that the generator of the signal localizes to parietal cortex. We agree here with the reviewer that scalp topography cannot be simply translated to anatomical origin of brain activity.

As indicated above, the resolution of the question is beyond the scope of this article, and the inclusion of the immediate response data has provided the necessary evidence to conclude that the observed CPP potential is consistent with a decision variable that is integrating sensory evidence. The resolution of the question of the different topographies will be separately investigated subsequently.

*But it could also be a signal issue – maybe the polynomial detrending has removed the actual decision signal from centro-parietal sites? "nt_detrend" probably has been used exclusively for datasets with relatively fast responses, and not for tasks where the potentials are very slow-moving like the current one – what happens to the signals when this step is removed, or replaced by a first-order (linear) detrend? It is possible in principle that strong auditory-evoked negative activity more frontally causes the positive focus to shift back on the scalp, but this seems an unlikely explanation given that the CPP/P300 have been measured with the same centro-parietal topographies for many auditory tasks including ones containing continuous audio streams (O'Connell et al., 2012).*

We have performed two tests to assess the potential influence of nt-detrend onto the recovery of slow potentials.

1) Reanalysis of the delayed response data with a classical high-pass filter (Matlab: filtfilt, 0.1 Hz, 15th order, 50 dB attenuation in the stop band) on the new dataset. The results were quite similar although obviously not identical (Figure 11).

2) Analysis of the immediate response EEG data with a classical high-pass filter (same setting as above). The results of this analysis are provided below, and show very similar results as for the filtering with nt_detrend. Especially, slow potentials are not removed to a larger degree than for high-pass filtering (see below, Figure 5—figure supplement 2).

Author response image 3.Recreation of Figure 5 for the delayed paradigm with a classical highpass filter, same caption (compare to Figure 10).**DOI:**
http://dx.doi.org/10.7554/eLife.24910.022

*Abstract: "in parietal cortex" should definitely be replaced with "over parieto-occipital scalp". This is EEG.*

We changed this description.

[Editors’ note: the author responses to the re-review follow.]

*Reviewer #1:*

*In this paper, Boubenec and colleagues demonstrate compelling psychophysical and electrophysiological correlates of statistical estimation mechanisms underlying naturalistic acoustic change detection, and provide both an abstract and physiologically-based computational model to explain their data. This is a revision of a previous submission, based on new EEG data recorded during the exact same task as was physiologically characterized and modeled, and the results are now more definitive and highly interesting, including stronger evidence for accumulation (steeper buildup for stronger changes) and additional effects on peak amplitude of the neural index of accumulation. The concepts and hypotheses are now also laid out very clearly, with nice sections explaining predictions based on statistical estimation strategies. There are just a couple of substantive points that are important to clear up at this stage.*

*There are a couple of important checks to perform on the CPP amplitudes at response to verify the effects on this measure. First, the topography of the effect of change size on amplitude should be shown, around the time of response, e.g. the largest change size minus smallest – does it have a similar CPP topography? This would go some way to verifying that the amplitude effects are not caused by an overlapping, separate process (e.g. slow fronto-central negativity).*

We have performed this test subtracting the 140% (largest) and 50% (smallest) conditions from each other. The topography is shown now as an inset in Figure 5. The topography is naturally smaller and noisier (mostly introduced by the 50% condition with less hits), but otherwise shows the same shape, also peaking in similar locations. The color-scale is the same as in the main figure of Figure 5, although it would be an option to increase it for clarity.

*Also, since the baseline for the response-locked waveforms was 1000-1200ms prior to response, conditions with longer RTs are baselined relative to intervals further out into the stimulus; this is a potential confound because it could be that for lower change sizes, the hit trials are those in which the CPP waveform gets a head-start by accumulating noise or a false change, and so it has less of a distance to go to threshold when the actual change happens. Or given how long some RTs are for weaker changes, the "head-start" could be from genuine physical evidence! This would result in an amplitude decrease at response when using the current baselining regime, even in the absence of any change in bound. Therefore a pre-stimulus or a pre-change baseline should instead be used for the response-locked waveforms. It is important to see whether this changes the results. I would be fairly confident that the main effects hold up – the slope effects certainly will because baselining doesn't matter in that case. Nevertheless, these controls are important to rule out critical potential confounds.*

We agree that this choice of baseline is more sensible in general (for the Response plots). We have changed the analysis accordingly and replot the results in Figure 5 (this applies to panels B2, E2, and all panels subsequent to (>=) G). No qualitative changes were observed although the p-values and averages were changed slightly. Specifically, the significance for the amplitude in response decreased, however, still appears quite solid. The baselining description in Methods has been adapted accordingly, as follows:

“The epochs were baseline-corrected relative to a 150 ms interval prior to onset, and a 200 ms interval before change time for alignments to both change and response time”.

*Possible explanations for the decrease in amplitude of CPP across change times tend to be provided briefly in a scattered fashion. It is important to reiterate the lack of effect of RT here – if bound is collapsing over time, surely the CPP amplitude would decrease with RT as well as with change time?*

An excellent point to check! The results shown in Figure 6 show a tiny, non-significant decrease in the predicted direction (0.1μV). We checked separately in all change time bins, but the effect was not significant in either bin. However, it is worth considering the expectable effect size: The overall decrease in threshold as suggested from Figure 5 would be 4.3-3.0μV = 1.3μV over the period of 4s. The average difference in RT between early and late (as analyzed in Figure 6) would be ~0.5s (estimated from the Data in Figure 3). Hence the expectable effect size over this time-difference would be 1.4/(4/0.5) = ~0.16μV. The observed difference across conditions in Figure 6 is 0.1μV. We therefore conclude that as far as we can tell, the data are qualitatively consistent with this hypothesis, or at least not inconsistent. However, the expectable effect-size is small, so that one would need substantially more data to perform this control. This is now stated in the manuscript as follows:

“Although the time-dependence of CPP height could result in a decrease of CPP height for late versus early reaction times, we did not find any significant decrease in CPP height for late reaction times, which may be due to a rather small effect-size (Figure 6).”

*Further, as the authors point out themselves, if bound were collapsing over time, false alarm rate should increase – it seems that this point should be weaved into the discussion of this effect more than it is. In general, since several possible explanations are given for this effect, it might be better not to scatter them in various parts of the Results and Discussion, but rather go through them all in one coherent section in the discussion?*

*Around paragraph three of subsection “EEG recordings and the site of decision-making”the discussion is a bit muddled – it appears to both argue for and against a decreasing bound in the same breath. This could use clarifying.*

We have reduced the interpretation to a minimum in the Results and centralized all of the considerations surrounding the reduction in height over time in the Discussion (which is arguably a better place than the Results, while the legibility in the Results may have benefitted from it). We would like to mention that a decreasing threshold (applied to the neural response, rather than the stimulus) is not necessarily inconsistent with a flat false alarm rate. If the response potential also decreases in size for the same deviation in the stimulus over time, these two effects could balance each other. Such a decrease in response potential could be predicted on the basis that certain stimulus changes may appear unexpected in the beginning, but are later expected after sampling the statistics for some time.

*Finally on this point, have the authors considered other, simpler mechanisms for preventing the subject from clicking when the stimulus initially onsets? The work of Tobias Teichert (see e.g. review in 2014) and others have demonstrated that subjects may be in control of the timing of onset of decision processes (specifically evidence accumulation) – given there's a "dead zone" apparent in Figure 1 showing that subjects are not willing to respond until a certain time has elapsed, couldn't they be simply deferring the kick-off of their decision mechanisms? This is easily parameterized in a fixed delay, and would avoid the need to introduce dramatic time-dependence in the fixed mechanisms (e.g. lengthening of the slow estimator's time constant) to explain that aspect. The authors should at least discuss this possibility.*

We agree that this is an alternative for modeling the strategy at the beginning of the trials. The current choice of a time-constant for the divergence of the integration properties can be considered a soft variant of your suggestion of a fixed dead-time. Unless we make the transition to regular integration thereafter instantaneous, we would, however, have to introduce another parameter (i.e. the time-constant would be replaced by the dead-time, and another, faster time-constant). We think that this could be a good method to model specific 'shapes of reluctance to respond' at the beginning, however, to differentiate between them would require a separate experiment, for example, where for example the response window is changed in different blocks. We discuss these options now in the Discussion and propose a future experiment to differentiate between them.

*Figure 5: I think it might make more sense to have each signal (medial, CPP) in the rows, and the response-locked waveforms to the right of the stimulus locked ones, as is standard across neurophysiological decision research. The labels should also be consistent in nature – perhaps they should be "fronto-central" and "parietal?" – "medial" is not a good descriptor since both signals are medial.*

We had previously chosen this alignment, as it provided a match of the onset component (top row) with the corresponding topoplot. But we also acknowledge the more logical arrangement of time going from left to right. The plots have been rearranged accordingly. Also, 'CPP' has been renamed to 'parietal' (in the figure, however, kept as CPP in the rest of the manuscript), and 'medial' to 'central'. Since, the peak of the onset potential is taken in central electrode, we considered it slightly misleading to include 'fronto' in the name.

*Missing from Figure 5 main figure are the change-onset locked waveforms. Supplement 3 shows the auditory electrodes, but really both signals should be shown time locked to change onset for a complete view over the dynamics. The x-axes of the current figures can easily be cut to zoom in on the main dynamics (e.g. the current 500-ms baseline shown before stimulus onset need only be 100 ms), so there should be room to squeeze them in between. This is important given that in places there are references to what happens to the CPP in response to the change, not just prior to response.*

Good point. We have accordingly inserted the change-time locked potential in between the Onset and Response, and zoomed in on the onset plot. The axes widths were scaled correspondingly to keep the temporal relation between the plots. Correspondingly the Figure Supplement 3 has been removed. The parietal change-locked potentials exhibit similar dependencies on the stimulus parameters, however, the wide distribution of response times together with the response potentials leads to a more widely spread out potential shape than in our previous dataset. References to the change-time related plots have been inserted and all labels changed in text and captions.